# Learning World Models for Interactive Video Generation

**Taiye Chen**[1*] **Xun Hu**[2*] **Zihan Ding**[3*] **Chi Jin**[3†]
[1]Peking University  [2]University of Oxford  [3]Princeton University
yeyutaihan@stu.pku.edu.cn
{xh4421,zihand,chij}@princeton.edu

## Abstract

Foundational world models must be both interactive and preserve spatiotemporal coherence for effective future planning with action choices. However, present models for long video generation have limited inherent world modeling capabilities due to two main challenges: compounding errors and insufficient memory mechanisms. We enhance image-to-video models with interactive capabilities through additional action conditioning and autoregressive framework, and reveal that compounding error is inherently irreducible in autoregressive video generation, while insufficient memory mechanism leads to incoherence of world models. We propose video retrieval augmented generation (VRAG) with explicit global state conditioning, which significantly reduces long-term compounding errors and increases spatiotemporal consistency of world models. In contrast, naive autoregressive generation with extended context windows and retrieval-augmented generation prove less effective for video generation, primarily due to the limited in-context learning capabilities of current video models. Our work illuminates the fundamental challenges in video world models and establishes a comprehensive benchmark for improving video generation models with internal world modeling capabilities.
Project page: https://sites.google.com/view/vrag.

## 1 Introduction

Foundational world models capable of simulating future outcomes based on different actions are crucial for effective planning and decision-making [1, 2, 3]. To achieve this, these models must exhibit both interactivity, allowing for action conditioning, and spatiotemporal consistency over long horizons. While recent advancements in video generation, particularly diffusion models [4, 5, 6, 7], have shown promise, extending them to generate long, interactive, and consistent videos remains a significant challenge [8, 9, 10].

Autoregressive approaches [11, 12, 13, 14], which generate videos frame by frame or chunk by chunk conditioned on previous outputs, are a

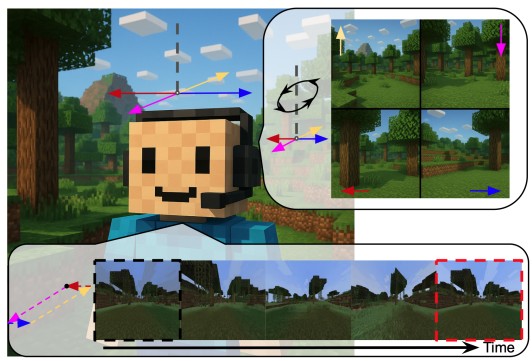

Figure 1: A world model possesses memory capabilities and enables faithful long-term future prediction by maintaining awareness of its environment and generating predictions based on the current state and actions. Example is in Minecraft game.

---

*Equal contribution.

†Corresponding Author

39th Conference on Neural Information Processing Systems (NeurIPS 2025).

natural fit for modeling long temporal dependencies and incorporating interactivity. However, these methods face significant challenges stemming from two fundamental, often coupled, limitations: **compounding errors** and **insufficient memory mechanisms**. Compounding errors arise as small inaccuracies in early predictions accumulate over time, leading to significant divergence from plausible future states. Our analysis suggests this may be inherent to current autoregressive paradigms. Insufficient memory mechanisms hinder the models' ability to maintain consistent object identities, spatial layouts, and world states over extended durations, resulting in inconsistent world models. These two issues often exacerbate one another, making long-term consistent generation difficult.

Inspired by the success of large language models (LLMs) [15][16] in handling long sequences, we investigate analogous techniques for video generation. Extending the context window, while potentially alleviating compounding errors to some degree, introduces substantial computational and memory overhead. More critically, we find that unlike LLMs, current video generation models exhibit weaker in-context learning capabilities, making longer context less effective in resolving fundamental consistency issues. Similarly, retrieval-augmented generation (RAG) [17][18], a powerful technique for incorporating external knowledge in LLMs, shows limited benefits in our experiments with video models. Neither static retrieval with heuristic sampling nor dynamic retrieval based on similarity search significantly improved world model consistency.

These findings suggest that implicitly learning world consistency solely from autoregressive prediction on pixel or latent representations is insufficient. We argue that explicit **global state conditioning** is necessary. Incorporating explicit representations like world maps, object states, or coordinate systems as conditioning information could provide the necessary grounding for generating consistent long-term interactive simulations.

Furthermore, evaluating the specific failure modes of long video generation demands appropriate metrics. Existing metrics often conflate the distinct issues of compounding errors and long-term consistency (memory faithfulness), providing a coupled assessment that obscures the underlying problems. To enable a clearer analysis, we advocate for and introduce a decoupled evaluation strategy by separately quantify the severity of compounding errors and the faithfulness of memory retrieval in long interactive video generation.

Our main contributions are: (1). We systematically decouple and analyze the challenges of compounding errors and insufficient memory in autoregressive video generation for interactive world modeling. (2). We propose video retrieval augmented generation (VRAG) with explicit global state conditioning, which significantly improves long-term spatiotemporal coherence and reduces compounding errors for interactive video generation. (3). We conduct a comprehensive comparison with various long-context methods adapted from LLM techniques, including position interpolation, neural memory augmentation, and historical frame retrieval, demonstrating their limited effectiveness due to the inherent weak in-context learning capabilities of video diffusion models. This work sheds light on the fundamental obstacles in building consistent, interactive video world models and provides a benchmark and evaluation framework for future research in this direction.

## 2 Related Works

**Video Diffusion Models** Diffusion generative modeling has significantly advanced the fields of image and video generation [19, 12, 20, 21, 22, 23, 24, 25, 26, 27, 28, 29]. Latent video diffusion models [21] operate on video tokens within a latent space derived from a variational auto-encoder (VAE) [30], building upon prior work in latent image diffusion models [31]. The Diffusion Transformer (DiT) [32] introduced the Transformer [33] backbone as an alternative to the previously prevalent U-Net architecture [23, 21, 22] in diffusion models.

**Long Video Generation** Autoregressive video generation [11, 12, 13, 14, 34, 35, 36, 37, 38, 39] represents a natural approach for long video synthesis by conditioning on preceding frames, drawing inspiration from successes in large language models. This can be implemented using techniques such as masked conditional video diffusion [40, 34] or Diffusion Forcing [41]. Diffusion Forcing introduces varying levels of random noise per frame to facilitate autoregressive generation conditioned on frames at inference time. Furthermore, the autoregressive framework naturally supports interactive world simulation by allowing action inputs at each step to influence future predictions. Nevertheless,

compounding errors remain a significant challenge in long video generation, particularly within the autoregressive paradigm, as will be discussed subsequently.

**Interactive Video World Models**  World models [1, 2, 3] are simulation systems designed to predict future trajectories based on the current state and chosen actions. Diffusion-based world models [42, 43, 10] facilitate the modeling of high-dimensional distributions, enabling high-fidelity prediction of diverse trajectories, even directly in pixel space. The Sora model [8] introduced the concept of leveraging video generation models as world simulators. Extending video generation models with interactive capabilities has led to promising applications in diverse domains, including game simulation like Genie [9], GameNGen [10], Oasis [44], Gamegen-x [45], The Matrix [37], Mineworld [46], GameFactory [47] and so on [43], autonomous driving [48], robotic manipulation [35, 49], and navigation [50]. While existing work on interactive video world models has made significant engineering advances, there remains a notable gap in systematically analyzing and addressing the fundamental challenges underlying long-term consistency and compounding errors.

A lack of spatiotemporal consistency is a primary bottleneck for developing internal world models using current video generation techniques. One line of research addressing this involves predicting the underlying 3D world structure like Genie2 [51], Aether [52], Gen3C [53] and others [54, 55, 56]; however, these approaches often suffer from lower resolution compared to direct video generation due to the complexity of 3D representations, exhibit limited interaction capabilities, and typically operate only within localized regions. Consequently, our work focuses on enhancing the consistency of video-based world models [10, 34, 57]. SlowFast-VGen [34] employs a dual-speed learning system to progressively trained LoRA modules for memory recall, utilizing semantic actions but offering limited interactivity. Concurrent work [57] explores interactive world simulation through the integration of supplementary memory blocks.

## 3 Methodology

### 3.1 Preliminary: Latent Video Diffusion Model

Video diffusion models have emerged as a powerful framework for video generation. We adopt a latent video diffusion model [21] that operates in a compressed latent space rather than pixel space for computational efficiency. Specifically, given an input video sequence $\boldsymbol{x} \in \mathbb{R}^{L \times H \times W \times 3}$, we first encode it into a latent representation $\boldsymbol{z} = \mathcal{E}(\boldsymbol{x})$ using a pretrained variational autoencoder (VAE). The forward process gradually adds Gaussian noise to the latent according to a variance schedule $\{\beta_t\}_{t=1}^{T}$:

$$q(\boldsymbol{z}_t|\boldsymbol{z}_{t-1}) = \mathcal{N}(\boldsymbol{z}_t; \sqrt{1-\beta_t}\boldsymbol{z}_{t-1}, \beta_t\mathbf{I}) \tag{1}$$

The model learns to reverse this process by predicting the noise $\boldsymbol{\epsilon}_\theta$ at each step:

$$\mathcal{L} = \mathbb{E}_{t,\boldsymbol{\epsilon},\boldsymbol{z}}[\|\boldsymbol{\epsilon} - \boldsymbol{\epsilon}_\theta(\boldsymbol{z}_t, t)\|_2^2] \tag{2}$$

where $\boldsymbol{z}_t = \sqrt{\bar{\alpha}_t}\boldsymbol{z}_0 + \sqrt{1-\bar{\alpha}_t}\boldsymbol{\epsilon}$ with $\boldsymbol{\epsilon} \sim \mathcal{N}(0, \mathbf{I})$.

At inference time, we can sample new videos by starting from random noise $\boldsymbol{z}_T \sim \mathcal{N}(0, \mathbf{I})$ and iteratively denoising:

$$\boldsymbol{z}_{t-1} = \frac{1}{\sqrt{\alpha_t}}(\boldsymbol{z}_t - \frac{\beta_t}{\sqrt{1-\bar{\alpha}_t}}\boldsymbol{\epsilon}_\theta(\boldsymbol{z}_t, t)) + \sigma_t\boldsymbol{\epsilon} \tag{3}$$

where $\alpha_t = 1 - \beta_t$ and $\bar{\alpha}_t = \prod_{s=1}^{t} \alpha_s$. The final latent sequence $\boldsymbol{z}_0$ is decoded back to pixel space using the decoder $\mathcal{D}$ to obtain the generated video.

### 3.2 Interactive Long Video Generation

To enable interactive long video generation conditioned on action sequences, we augment the base diffusion model with two techniques: (1) additional action condition with adaptive layer normalization (AdaLN), and (2) random frame noise for autoregressive modeling, as shown in diagram Fig. 2.

**Action Conditioning**  To enable interactive video generation conditioned on action sequences, we augment the base diffusion model with adaptive layer normalization (AdaLN). Given an action sequence $\boldsymbol{a} \in \mathbb{R}^{L \times A}$ where $A$ is the action dimension, we first embed it into a latent space using a

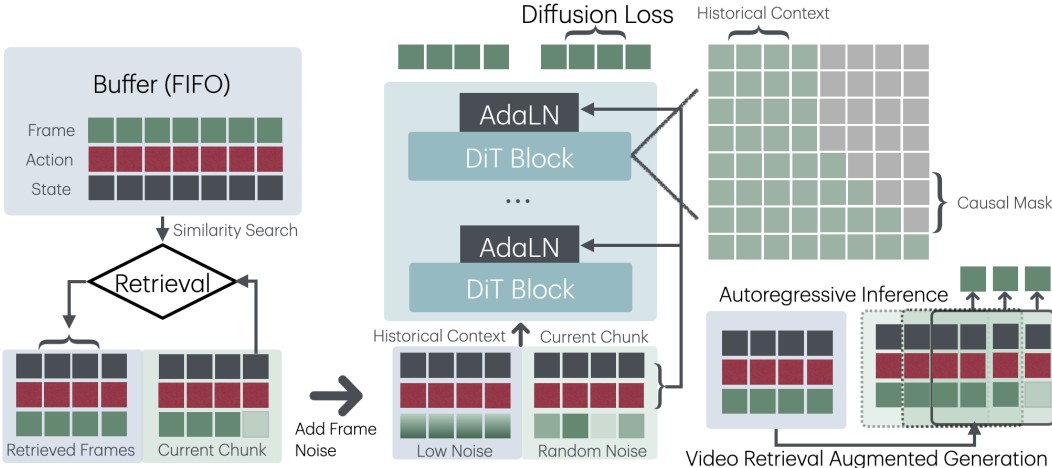

Figure 2: Overview of our VRAG framework for interactive video generation. The framework incorporates global state conditioning and memory retrieval mechanisms to ensure spatiotemporal consistency and mitigate error accumulation. During both training and inference, retrieved memory serves as context for joint self-attention in spatiotemporal DiT blocks. The model employs per-frame noise injection during training to facilitate autoregressive sampling at inference time.

learnable embedding layer: $e_a = \text{Embed}(\boldsymbol{a}) \in \mathbb{R}^{L \times D_e}$ where $D_e$ is the embedding dimension. For each normalization layer in the diffusion model, we learn action-dependent scale and shift parameters through linear projections: $\gamma_a = e_a W_\gamma + b_\gamma \in \mathbb{R}^{L \times D_h}, \beta_a = e_a W_\beta + b_\beta \in \mathbb{R}^{L \times D_h}$, where $D_h$ matches the hidden dimension of the feature maps. We have $\text{AdaLN}(h) = \gamma_a \odot \text{LayerNorm}(h) + \beta_a$, where $h \in \mathbb{R}^{L \times D_h}$ represents the intermediate feature maps and $\odot$ denotes dot production.

**Autoregressive Video Generation**   To enable long video generation, we adopt an autoregressive approach where we generate frames sequentially. At each step, we condition on a fixed-length context window $L_c$ of previously generated frames. However, naive autoregressive generation with teacher forcing can suffer from large compounding errors where mistakes accumulate over time. We apply the Diffusion Forcing [41] technique during training.

Specifically, during training, we randomly add noise to each frame in the entire input video sequence according to the diffusion schedule: $z_t^i = \sqrt{\bar{\alpha}_t} z_0^i + \sqrt{1 - \bar{\alpha}_t} \epsilon^i, \epsilon^i \sim \mathcal{N}(0, \mathbf{I})$, where $z_t^i$ represents the noised latent of the $i$-th frame. This forces the model to be robust to noise in the conditioning frames and prevents it from relying too heavily on the context. With above two techniques, the training objective for action-conditioned autoregressive video models become:

$$\mathcal{L}_{\text{DF}} = \mathbb{E}_{[t],\boldsymbol{\epsilon},\mathbf{z},a}[\|\boldsymbol{\epsilon}_{[t]} - \boldsymbol{\epsilon}_\theta(\mathbf{z}_{[t]}, [t], \boldsymbol{a})\|_2^2], \quad \boldsymbol{\epsilon}_{[t]} = \{\epsilon_t^i\}_{i=1}^L, \mathbf{z}_{[t]} = \{z_t^i\}_{i=1}^L \quad (4)$$

where $[t]$ is vector of $L$ timesteps with different $t \in [T]$ for each frame. The noise prediction model $\boldsymbol{\epsilon}_\theta$ conditioned on both the action sequence $\boldsymbol{a}$ and noised frames $\mathbf{z}_t$.

**Architecture**   We apply diffusion transformer (DiT) for video generation modeling. We adopt spatiotemporal DiT block with separate spatial and temporal attention modules. Rotary Position Embedding (RoPE) [58] is applied for both attention modules, and temporal attention is implemented with causal masking.

### 3.3   Retrieval Augmented Video World Model with Global State

While the vanilla model in Sec. 3.2 provides a foundation for interactive video generation, it lacks robust mechanisms for maintaining long-term consistency and world model coherence. To address these limitations, we integrate memory retrieval and context enhancement with inspiration from LLMs, and incorporate video-specific approaches such as historical frame buffer and global state conditioning. These enhancements enable more consistent and coherent autoregressive video generation by providing the model with better access to historical context and spatial awareness.

**Global State Conditioning** To enhance spatial consistency in video generation, we incorporate global state information—specifically the character's current coordinates and pose—as an additional conditioning signal. The global state vector $s \in \mathbb{R}^S$ consists of two key components: $s_{\text{pos}}$ representing 3D position coordinates and $s_{\text{ori}}$ capturing orientation angles. Given an action sequence $\boldsymbol{a} \in \mathbb{R}^{L \times A}$ and the global state sequence $\boldsymbol{s} \in \mathbb{R}^{L \times S}$, both are transformed by a learnable embedding layer, $e_c = \text{Embed}_c(\boldsymbol{a}, \boldsymbol{s})$, to produce conditioning features. These features are then fed into AdaLN layers within the diffusion model. This mechanism allows the model to modulate its generation process, adapting to both the input actions and the character's spatial context, thereby improving overall coherence.

**Video Retrieval Augmented Generation (VRAG)** Beyond global state conditioning, we propose memory retrieval augmented generation to enhance the model's ability to leverage historical context while maintaining temporal coherence, namely video retrieval augmented generation (VRAG). For VRAG, we combine the concatenated historical and current frames with their corresponding action sequences $\tilde{\boldsymbol{a}} \in \mathbb{R}^{L \times A}$ and global state sequences $\tilde{\boldsymbol{s}} = [\boldsymbol{s}_{\text{hist}}, \boldsymbol{s}] \in \mathbb{R}^{L \times S}$ as conditional inputs to the model. The historical frames are retrieved from a fixed-length buffer $\mathcal{B}$, which stores previously generated frames. The per-frame retrieval process is based on a heuristic sampling strategy, where we select the most relevant historical frames based on similarity search to concatenate with the current context. The similarity score based on global state is defined as:

$$r(\hat{s}) = f_{\text{sim}}(\hat{s} \odot w, s_{L-1} \odot w), \hat{s} \in \mathcal{B} \tag{5}$$

where $f_{\text{sim}}$ is a distance metric (e.g., Euclidean distance) between the history frame and the last frame to be predicted $s_{L-1}$, and $w \in \mathbb{R}^S$ is a weight vector that modulates the importance of different state components. The top $L_h$ most similar historical states and frames are selected and sorted to form the retrieved context. Unlike RAG in LLMs which leverages strong in-context learning capabilities, **video diffusion models exhibit weak in-context learning abilities, making direct inference with historical frames as context ineffective**, as demonstrated later in our experiments. To address this limitation, we propose VRAG training with key modifications to the standard RAG approach, enabling effective memory-augmented video generation.

During training, we retrieve historical frames $\mathbf{z}_{\text{hist}} \in \mathbb{R}^{L_h \times D}$ and concatenate them with the current context window $\mathbf{z} \in \mathbb{R}^{L_c \times D}$ to form the extended context $\tilde{\mathbf{z}} = [\mathbf{z}_{\text{hist}}, \mathbf{z}]$. For effective VRAG, we make several key modifications: (1). To distinguish retrieved frames from normal context frames, we modify the RoPE embeddings by adding a temporal offset $\Delta t$ to the retrieved frames' position indices. (2). Additionally, we apply lower noise levels $\beta_{t'} < \beta_t$ to the retrieved frames $\mathbf{z}_{\text{hist}}$ to simulate partially denoised historical frames during inference. This enhances the robustness of the model with imperfect historical frames generated previously during the autoregressive process. The model is trained to denoise for the entire context $\tilde{\mathbf{z}}$ including both retrieved and current frames. (3). To ensure the model focuses on denoising the current context while leveraging historical information, we mask the diffusion loss $\mathcal{L}_{\text{DF}}$ for retrieved frames. (4). Furthermore, for retrieved frames, we only condition on their global states $\boldsymbol{s}_{\text{hist}} \in \mathbb{R}^{L_h \times S}$, masking out action conditions $\boldsymbol{a}_{\text{hist}} \in \mathbb{R}^{L_h \times A}$ to avoid temporal discontinuity in action sequences. This selective conditioning approach helps maintain spatial consistency while preventing action-related artifacts from propagating through the generation process. Overall, the training objective of VRAG on diffusion models is defined as:

$$\mathcal{L}_{\text{VRAG}} = \mathbb{E}_{[t],[t'],\boldsymbol{\epsilon},\tilde{\mathbf{z}},a,s}[\|\boldsymbol{\epsilon}_t - \boldsymbol{\epsilon}_\theta(\tilde{\mathbf{z}}_{\tilde{t}}, \tilde{t}, \tilde{\boldsymbol{a}}, \tilde{\boldsymbol{s}})\|_2^2 \odot \mathbf{m}], \tag{6}$$

$$\tilde{\mathbf{z}}_{\tilde{t}} = [\mathbf{z}_{\text{hist},[t']}, \mathbf{z}_{[t]}], \quad \tilde{\boldsymbol{a}} = [\varnothing_{L_h}, \boldsymbol{a}], \quad \tilde{\boldsymbol{s}} = [\boldsymbol{s}_{\text{hist}}, \boldsymbol{s}], \quad \mathbf{m} = [\mathbf{0}_{L_h}, \mathbf{1}_{L_c}], \tag{7}$$

where $\tilde{t}$ is a concatenation of $[t']$ and $[t]$, with $t' < t$ and $t', t \in [T]$.

### 3.4 Long-context Extension Baselines

To investigate whether established long-context extension techniques from LLMs can effectively enhance video generation models, we design three complementary approaches that leverage either explicit frame context or neural memory hidden states, based on vanilla models in Sec. 3.2. These methods serve as baseline comparisons to our main approach, specifically targeting the model's ability to maintain spatial coherence and temporal consistency in long video generation. Through these baselines, we aim to verify the in-context learning capabilities of video diffusion models and assess their effectiveness in handling extended sequences.

**Long-context Enhancement**   We extend the temporal context window using YaRN [59] modification for RoPE in temporal attention. RoPE encodes relative positions via complex-valued rotations, where the inner product between query $\mathbf{q}_m$ and key $\mathbf{k}_n$ depends on relative distance $(m - n)$. YaRN extends the context window by applying a frequency transformation to the rotary position embeddings. This transformation scales the rotation angles in a way that preserves the relative positioning information while allowing the model to handle longer video sequences, after small-scale fine-tuning on longer video clips.

**Frame Retrieval from History Buffer**   We implement a fixed-length buffer $\mathcal{B}$ storing historical latent frames with a heuristic sampling strategy. The buffer is partitioned into $N_S = 5$ exponentially decreasing segments $G_j$, where $L_j = L_1 \cdot \alpha^{j-1}$. From each segment $G_j$, we sample $k$ frames to form subset $F_j$. The retrieved memory $\mathbf{z}_{\text{mem}} = [F_1, \ldots, F_{N_S}]$ is concatenated with current frame window $\mathbf{z}$ as additional context: $\tilde{\mathbf{z}} = [\mathbf{z}_{\text{mem}}, \mathbf{z}]$, which is then passed into the spatiotemporal DiT blocks. This design ensures higher sampling density for recent frames, emphasizing recent visual information while maintaining access to historical context for temporal consistency.

**Neural Memory Augmented Attention**   Instead of using explicit frames as context in above two methods, we explore a neural memory mechanism to store and retrieve hidden states. This approach is inspired by the success of Infini-attention [60] in LLMs, which utilizes a compressed memory representation to enhance attention mechanisms. The model processes video in overlapping segments to maintain temporal continuity. For each video segment $\mathbf{z}_s$, we compute query $\mathbf{q}_s$, key $\mathbf{k}_s$ and value $\mathbf{v}_s$ matrices. The model retrieves hidden state $\mathbf{A}_{\text{mem}}$ from compressive memory $\mathbf{M}_{s-1}$: $\mathbf{A}_{\text{mem}} = \frac{\sigma(\mathbf{q}_s)\mathbf{M}_{s-1}}{\sigma(\mathbf{q}_s)\mathbf{n}_{s-1}}$. Memory $\mathbf{M}_{s-1}$ and normalization vector $\mathbf{n}_{s-1}$ are then updated. The final attention output combines retrieved hidden state $\mathbf{A}_{\text{mem}}$ and standard attention using learnable gating to maintain visual consistency across the long video sequence.

**Frame Pack**   As another baseline, we follow the Frame Pack [61] to compress historical frames as context. Three input compression kernels with different kernel sizes-(2, 4, 4), (4, 8, 8), and (8, 16, 16)-are employed to condense the historical frames into a fixed-length context. This approach essentially achieves frame compression through importance sampling with recency bias, which enables a larger field of view while maintaining lower computational costs. However, the prioritization of most recent frames can be suboptimal in many cases for long video generation especially when considering the memory issue. Our VRAG based on frame relevance provides theoretically better historical information retrieval. Moreover, our method is actually orthogonal to the frame compression technique in Frame Pack. We leave the combined methods as future work.

More details of the above methods can be found in the supplementary material.

## 4   Experiments

### 4.1   Datasets and Evaluation Protocol

For training, we collected 1000 long Minecraft gameplay videos (17 hours total) using MineRL [62]. All videos have a fixed resolution of 640×360 pixels. Each sequence spans 1200 frames, annotated with action vectors (forward/backward movement, jumping, camera rotation) and world coordinates (x, y, z positions and yaw angle).

For evaluation, we assembled two distinct test sets: (1) for compounding error evaluation, we use 20 long videos of 1200 frames with randomized actions and locations, and (2) for world coherence, we use 60 carefully curated 300-frame video sequences designed to systematically assess spatiotemporal consistency. These curated sequences feature controlled motion patterns including in-place rotation, direction reversal, and circular trajectory following. The first 100 frames of each sequence serve as initialization buffer for methods requiring buffer frames or are excluded from evaluation for others. Each model autoregressively generates next single frame with stride 1 until the desired length.

We evaluate the models against ground-truth test sets using several metrics: Structural Similarity Index (SSIM) [63] to measure spatial consistency, Peak Signal-to-Noise Ratio (PSNR) for pixel-level reconstruction quality, Learned Perceptual Image Patch Similarity (LPIPS) [64] to assess perceptual similarity. For the compounding error evaluation, we find SSIM more accurately reflect the faithfulness of frames over long sequences.

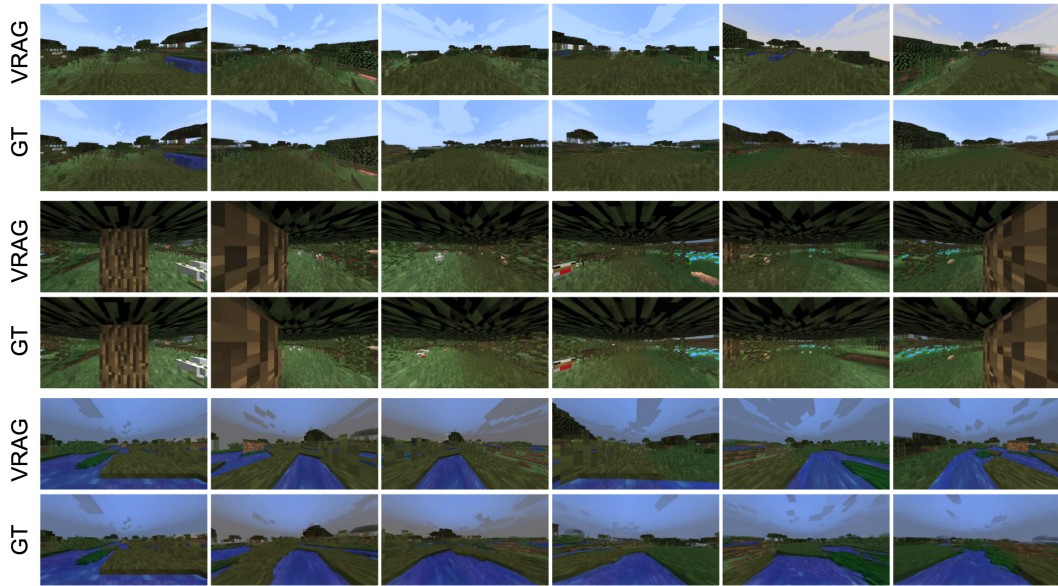

Figure 3: Visual comparison of VRAG with ground truth videos on world coherence evaluation. With 100 initial frames as history buffer, VRAG predicts 200 subsequent frames.

## 4.2 Training Details

A consistent window size of 20 frames is applied for both model training and evaluation for fair comparison. For vanilla Diffusion Forcing, we additionally train a variant with window sizes of 10 frame for context length evaluation. For our VRAG method, we combine 10 retrieved frames with 10 current frames for both training and inference. We represent the agent's state using a global state vector $s = [x, y, z, \text{yaw}]$ during training, which can be extended to incorporate a full 3D pose representation when needed. To facilitate training convergence, these values are normalized relative to the initial state, thereby

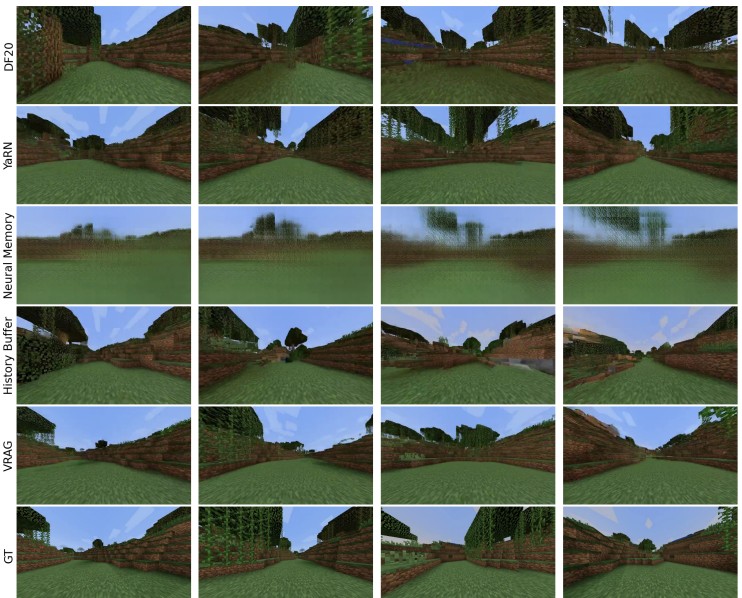

Figure 4: Visual comparison of different methods, evaluated for world coherence analysis.

reducing the complexity of the diffusion process. The YaRN implementation extends the vanilla model (window size 20) by replacing position embeddings with YaRN and stretching factor 4, followed by fine-tuning for $10^4$ steps on 80-frame sequences. During evaluation of Yarn, we use a 40-frame window. The Infini-attention with neural memory employs a sliding window size 20 and stride 10, using the first 10 frames for memory state updates and the last 10 for local attention computation. The History Buffer method maintains a 124-frame buffer partitioned into 5 exponentially decreasing segments ($L_1 = 2, \alpha = 2$), sampling 2 frames per segment to form 10 historical frames that are concatenated with the 10 current frames. All models are trained for 3 epochs on the dataset, with a batch size of 32 across 8 A100 GPUs.

## 4.3 World Coherence Results

We investigate the spatiotemporal consistency of internal world models by evaluating the predicted videos given initial frames and action sequences. As visualized in Fig. 4, our VRAG provides an effective approach to enhance the model's ability to leverage historical context for improving world coherence. Fig. 3 shows more visual comparison of VRAG with ground truth videos. We evaluate the world coherence of different methods using multiple metrics. Fig. 5 shows the SSIM scores over time, while Tab. 1 presents a comprehensive comparison across all metrics. Our VRAG method achieves the best performance across all metrics, demonstrating its superior ability to maintain world coherence in generated videos. Our experimental results demonstrate that expanding the window size from 10 to 20 frames in the baseline DF model improves world coherence, indicating that longer context windows enhance consistency. However, further context extension using YaRN shows no improvement over the vanilla DF model. This suggests that YaRN's context extension capabilities, while effective in language models, do not transfer effectively to video generation for maintaining world coherence. Similarly, the History Buffer method fails to effectively utilize historical frames for spatiotemporal consistency without explicit in-context training. These findings from both YaRN and History Buffer approaches reveal that video diffusion models at the current scale possess limited in-context learning capabilities, preventing them from effectively leveraging historical frames for maintaining long-term consistency. The Neural Memory method performs poorly due to its instability in model training.

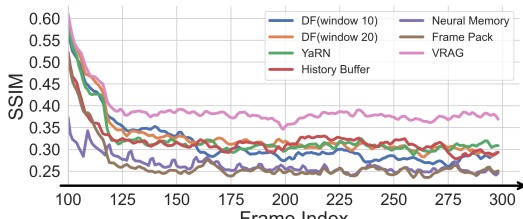

Figure 5: SSIM scores over time for different methods on world coherence evaluation.

| Method | SSIM ↑ | PSNR ↑ | LPIPS ↓ |
|---|---|---|---|
| DF (window 10) | 0.455 | 16.161 | 0.509 |
| DF (window 20) | 0.466 | 16.643 | 0.538 |
| YaRN | 0.462 | 16.567 | 0.532 |
| History Buffer | 0.459 | 16.922 | 0.543 |
| Frame Pack | 0.421 | 16.372 | 0.574 |
| **VRAG** | **0.506** | **17.097** | **0.506** |

Table 1: Quantitative comparison of world coherence across different methods, evaluated on videos with 300 frames.

## 4.4 Compounding Error Results

We evaluate the compounding error in long video generation across different methods using the SSIM metric. As shown in Fig. 7 and Tab. 2, our VRAG method achieves superior performance with an SSIM score of 0.349, demonstrating better structural similarity preservation compared to baseline methods. Increasing the window size in DF from 10 to 20 frames improves SSIM, indicating that longer context helps mitigate compounding errors. However, this improvement is still inferior to

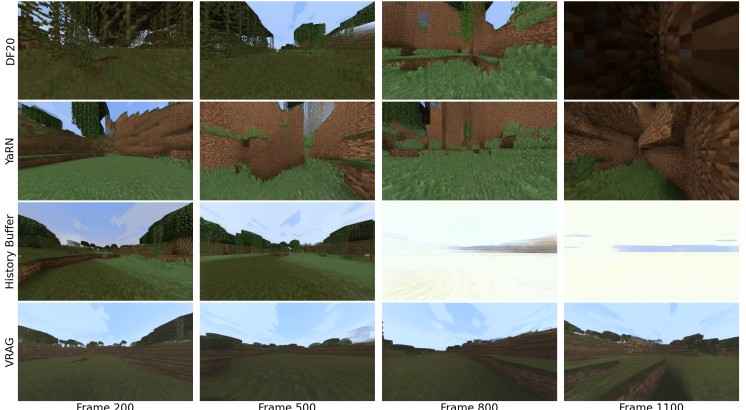

Figure 6: Visual comparison of long-term video prediction (1200 frames) across different methods, evaluated for compounding error analysis.

VRAG's performance, suggesting that our retrieval-augmented approach provides more effective long-term consistency. As visualized in Fig. 6, our VRAG method generates more coherent and consistent frames over long sequences, while other methods exhibit noticeable artifacts and inconsistencies. The History Buffer method performs poorly, with an SSIM score of 0.188, indicating that naive historical frame retrieval without effective in-context training fails to maintain long-term

consistency. Given its limited performance in the world coherence experiments (Sec. 4.3), we exclude the Neural Memory method from visualization in this longer video prediction visualization.

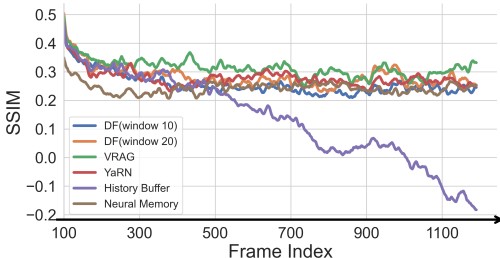

Figure 7: SSIM scores over time for compounding error evaluation

| Method | SSIM ↑ |
|---|---|
| DF (window 10) | 0.297 |
| DF (window 20) | 0.321 |
| YaRN | 0.316 |
| History Buffer | 0.188 |
| Neural Memory | 0.283 |
| **VRAG** | **0.349** |

Table 2: Average SSIM scores across all frames in compounding error evaluation

## 4.5 VBench Evaluation

We evaluate the long video generation with five Video Quality metrics in VBench [65] for generated videos in Sec. 4.4. The evaluation results on VBench (higher is better) are shown in the Tab. 3. As demonstrated in the results, our method outperforms all other baselines across all metrics in both temporal quality and video frame quality. The Neural Memory baseline has Aesthetic Quality 0.343 and Imaging Quality score 0.3597 respectively, which are significantly lower than other baselines, therefore not listed here. Our VRAG shows better temporal consistency compared with all baseline methods, and the high video frame quality indicates the results are not over-smoothed.

## 4.6 Extension: Real World Setting

| Method | Background Consistency | Temporal Flickering | Motion Smoothness | Aesthetic Quality | Imaging Quality |
|---|---|---|---|---|---|
| DF20 | 0.9668 | 0.9485 | 0.9582 | 0.5272 | 0.6058 |
| YaRN | 0.9686 | 0.9401 | 0.9523 | 0.5252 | 0.6323 |
| History Buffer | 0.9664 | 0.9475 | 0.9579 | 0.5167 | 0.6253 |
| VRAG | **0.9686** | **0.9511** | **0.9603** | **0.5295** | **0.6444** |

Table 3: Evaluation results on five Video Quality metrics in VBench.

| Metric | DFoT | VRAG |
|---|---|---|
| SSIM ↑ | 0.4436 | **0.9116** |
| PSNR ↑ | 13.03 | **32.21** |
| LPIPS ↓ | 0.4469 | **0.1146** |
| FVD ↓ | 337.5 | **221** |

Table 4: Quantitative comparison on RealEstate10K dataset.

We conduct additional experiments in real-world setting beyond Minecraft simulation to show generalization of our approach. Specifically, following the experimental setup of Diffusion Forcing Transformer (DFoT) [66], our VRAG model is initialized from pre-trained DFoT and finetuned on the RealEstate10K dataset [67] with additionally retrieved historical context, as described in Sec. 3.3.

After fine-tuning for just 2 epochs (10% of the original training steps), our method significantly outperforms the DFoT baseline in terms of memorization capability. The visualized frames are presented in Fig. 8 and quantitative results are summarized in Tab. 4. The results effectively demonstrate the generalization of our approach beyond Minecraft for solving the memory issue in long video prediction.

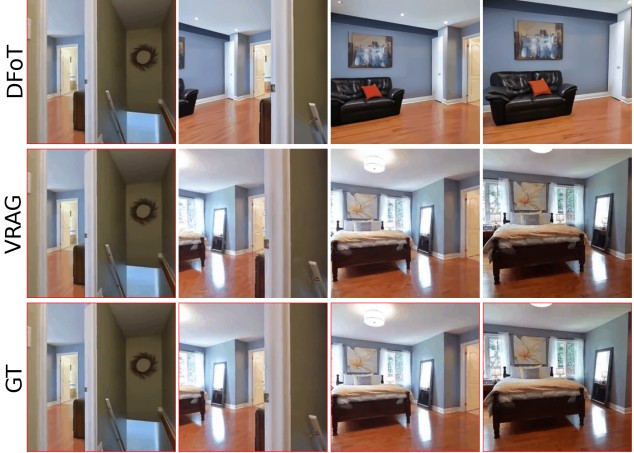

Figure 8: Visualized video frames on RealEstate10K dataset. Red blocks indicate the ground-truth frames.

## 4.7 Ablation: Memory and Training of VRAG

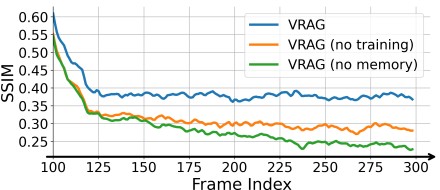

Figure 9: Comparison of SSIM scores over time for VRAG variants.

| Method | SSIM ↑ | PSNR ↑ | LPIPS ↓ |
|---|---|---|---|
| **VRAG** | **0.506** | **17.097** | **0.506** |
| VRAG (no training) | 0.455 | 16.670 | 0.528 |
| VRAG (no memory) | 0.436 | 16.372 | 0.547 |

Table 5: Ablation study of VRAG components. We compare the full model with variants that remove either the memory component (additional global state conditioning only) or training component (in-context learning only).

We ablate the key designs for VRAG methods, including the memory and training components. The ablation results are shown in Fig. 9 and Tab. 5. We compare the full VRAG model with two variants: (1) VRAG without the memory component, which only uses additional global state conditioning, and (2) VRAG without the training component, i.e., vanilla model with retrieval augmented generation for in-context learning at inference. The ablation study is conducted on the world coherence evaluation dataset.

The ablation results reveal several key insights about VRAG components. First, removing the memory component leads to the largest performance drop across all metrics, with SSIM decreasing by 13.8% and LPIPS increasing by 8.1%. This demonstrates that the memory mechanism is crucial for maintaining spatiotemporal consistency and quality. Second, removing the training component also causes significant degradation, with SSIM dropping by 10.1% and LPIPS increasing by 4.3%, highlighting the weak capabilities of in-context learning for current video models. The full VRAG model achieves the best performance across all metrics, showing that both components work synergistically to improve video generation quality.

# 5 Conclusions and Discussions

In conclusion, VRAG tackles the fundamental challenge of maintaining long-term consistency in interactive video world models through an innovative combination of memory retrieval-augmented generation and global state conditioning. By maintaining a buffer of past frames associated with spatial information, VRAG effectively recalls relevant context and preserves coherent dynamics across extended sequences. Its memory mechanism with explicit in-context training process substantially mitigates compounding errors and improves spatiotemporal consistency. Extensive experiments on long-horizon interactive tasks demonstrate the superior performance of VRAG over both long-context and memory-based baselines, establishing a scalable framework for faithful video-based world modeling. Notably, we discovered that context enhancement techniques from LLMs fail to transfer effectively to the video generation domain, even with shared transformer backbones, due to the inherent limitations of in-context learning capabilities for video models. This finding underscores the critical importance of VRAG's in-context training approach. We hope our work will inspire further exploration into memory retrieval mechanisms for long video generation and interactive simulation.

**Limitaitons.** We acknowledge the current computational limitations preventing effective scaling to longer sequences or larger architectures. GPU memory constraints severely restricted memory buffer size and training sequence length, potentially impacting long-horizon consistency and model performance. The higher computational cost of memory retrieval-augmented generation may further limit deployment in resource-constrained settings such as edge devices. Future work could explore more efficient memory mechanisms, adaptive optimization strategies, and hardware-aware algorithms.

**Broader Impacts.** We acknowledge serious ethical concerns regarding the potential misuse of such technology for creating highly convincing misleading or manipulated video content in games or simulation systems. We strongly encourage responsible development and deployment of video generation technologies, with appropriate technical and ethical safeguards, clear accountability frameworks, and transparency measures in place to mitigate risks.

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

# Appendix

## A  Baseline Method Details

For the baseline methods in Sec. 3.4, we implemented the following techniques to enhance the temporal context window of our video generation model.

**Long-context Enhancement**  To extend the temporal context window of our video generation model, we apply the YaRN [59] modification for ROPE in temporal attention module for improved extrapolation. RoPE encodes relative position via complex-valued rotations, such that the inner product between the $m$-th query $\mathbf{q}_m$ and $n$-th key $\mathbf{k}_n$ depends only on the relative distance $(m - n)$:

$$\langle \mathbf{q}_m, \mathbf{k}_n \rangle = \langle f_{\mathbf{W}_q}(\mathbf{z}_m, m), f_{\mathbf{W}_k}(\mathbf{z}_n, n) \rangle_{\mathbb{R}} \tag{8}$$

$$= \mathrm{Re}\left( \langle (\mathbf{W}_q \mathbf{z}_m) e^{im\theta}, (\mathbf{W}_k \mathbf{z}_n) e^{in\theta} \rangle \right) \tag{9}$$

$$= \mathrm{Re}\left( (\mathbf{W}_q \mathbf{z}_m)(\mathbf{W}_k \mathbf{z}_n)^* \cdot e^{i(m-n)\theta} \right) \tag{10}$$

$$= g(\mathbf{z}_m, \mathbf{z}_n, m - n) \tag{11}$$

where $\mathrm{Re}[\cdot]$ is real part of complex values and $(\cdot)^*$ represents conjugate of complex numbers, $\mathbf{z}_m, \mathbf{z}_n \in \mathbb{R}^D$ are input vectors, $\mathbf{W}_q, \mathbf{W}_k$ are learned projections, and $\theta \in \mathbb{R}^D$ encodes rotation frequencies per dimension: $\theta_d = b^{-2d/D}$, with $b = 10000$.

YaRN modifies modifies the rotated input vector $f_{\mathbf{W}}(\mathbf{z}_m, m, \theta_d)$ by applying a frequency transformation:

$$f'_{\mathbf{W}}(\mathbf{z}_m, m, \theta_d) = f_{\mathbf{W}}(\mathbf{z}_m, g(m), h(\theta_d)) \tag{12}$$

with $g(m) = m$ and frequency warping function:

$$h(\theta_d) = (1 - \gamma(r_d)) \cdot \frac{\theta_d}{s} + \gamma(r_d) \cdot \theta_d \tag{13}$$

Here, $s$ is a stretching factor and $r_d = L_c/\lambda_d$ is the context-to-wavelength ratio with $\lambda_d = 2\pi/\theta_d = 2\pi \left(b'\right)^{2d/D}$ and $b' = bs^{\frac{D}{D-2}}$. The ramp function $\gamma(\cdot)$ interpolates low-frequency dimensions to improve extrapolation while preserving high-frequency components.

**Frame Retrieval from History Buffer**  We also experimented with a fixed-length buffer $\mathcal{B}$ that stores a history of previously generated latent frames, employing a heuristic sampling strategy for retrieval. Following [68], this strategy involves partitioning $\mathcal{B}$ into $N_S = 5$ segments $G_j$ for $j \in \{1, \ldots, N_S\}$, ordered from oldest ($G_1$) to most recent ($G_{N_S}$). The total number of frames in the buffer is $N_B = \sum_{j=1}^{N_S} |G_j|$. The lengths of these segments, $L_j = |G_j|$, decrease exponentially (e.g., $L_j = L_1 \cdot \alpha^{j-1}$ for a base $\alpha < 1$ and $L_1$ being the length of the oldest segment $G_1$), ensuring that more recent segments are shorter. From each segment $G_j$, $k$ frames are randomly sampled to form a subset $F_j \subseteq G_j$ (where $|F_j| = k$). The retrieved memory $\mathbf{z}_{\text{mem}}$ is constructed as the concatenation of these sampled frames, $\mathbf{z}_{\text{mem}} = [F_1, F_2, \ldots, F_{N_S}]$, totaling $N_S \cdot k$ frames. This design with recency bias implies that the sampling density $k/L_j$ is higher for more recent segments, thereby placing greater emphasis on recent information. This retrieved information $\mathbf{z}_{\text{mem}}$ is concatenated with current frame window $\mathbf{z}$ along temporal dimension as additional context: $\tilde{\mathbf{z}} = [\mathbf{z}_{\text{mem}}, \mathbf{z}]$, which is then passed as input to the spatiotemporal DiT blocks, enabling the model to jointly attend to both recent and historical frames.

**Neural Memory Augmentation**  To extend video generation capabilities to longer sequences beyond a fixed attention window while retaining memory of past scenes, we adapt Infini-attention [60] as a neural memory mechanism for our video diffusion model. Infini-attention is a recurrent mechanism that augments standard dot-product attention (local context) with a compressed summary of past context (global context) stored in an evolving memory.

The model processes the video in segments using a sliding window. To maintain the high degree of temporal continuity crucial for video generation, we employ overlapping segments. This is a modification from the original Infini-attention, which typically processes non-overlapping segments. The input latent video segment $\mathbf{z}_s \in \mathbb{R}^{N \times D}$ ($s$ is segment index) is processed to derive query $\mathbf{q}_s$,

key $\mathbf{k}_s$ and value $\mathbf{v}_s$ matrices using standard attention mechanisms. Key-value pairs from processed segments are incrementally summarized and stored in a compressive memory $\mathbf{M}$, which can be efficiently queried by subsequent segments using their query vectors. After each slide, the model first retrieves a hidden state $\mathbf{A}_{\text{mem}}$ by querying the compressive memory $\mathbf{M}_{s-1}$:

$$\mathbf{A}_{\text{mem}} = \frac{\sigma(\mathbf{q}_s)\mathbf{M}_{s-1}}{\sigma(\mathbf{q}_s)\mathbf{n}_{s-1}} \tag{14}$$

where $\sigma(\cdot)$ is an element-wise nonlinear activation function (e.g., $\text{ELU}(\cdot) + 1$) and $\mathbf{n}_{s-1}$ is a normalization vector (accumulated up to segment $s-1$).

Next, the compressive memory $\mathbf{M}_s$ and normalization vector $\mathbf{n}_s$ are updated using the KV entries of the current segment $s$:

$$\mathbf{M}_s = \mathbf{M}_{s-1} + \sigma(\mathbf{k}_s)^T \left( \mathbf{v}_s - \frac{\sigma(\mathbf{k}_s)\mathbf{M}_{s-1}}{\sigma(\mathbf{k}_s)\mathbf{n}_{s-1}} \right)$$
$$\mathbf{n}_s = \mathbf{n}_{s-1} + \sigma(\mathbf{k}_s)^T \mathbf{1}_N \tag{15}$$

Here, $N$ is the length of the current segment $s$. $\sigma(\cdot)$ is applied element-wise, and $\mathbf{1}_N$ is an $N \times 1$ vector of ones.

The final attention output for segment $s$, denoted $\mathbf{A}_s$, combines the standard dot-product attention output $\mathbf{A}_{\text{local}}$ (local context from the current segment) with the retrieved memory state $\mathbf{A}_{\text{mem}}$ (global context from past segments) using a learnable gating scalar $\beta \in \mathbb{R}$:

$$\mathbf{A}_s = \text{sigmoid}(\beta) \odot \mathbf{A}_{\text{mem}} + (1 - \text{sigmoid}(\beta)) \odot \mathbf{A}_{\text{local}} \tag{16}$$

As in standard multi-head attention, a final linear projection is applied to $\mathbf{A}_s$ to produce the output of the Infini-attention layer.

## B  Implementation Details

The VAE compresses each input frame of size $3 \times 640 \times 360$ into a latent representation of size $16 \times 32 \times 18$ before processing by the diffusion model. All diffusion models employ a hidden size of 1024 and depth of 16, with one temporal and one spatial attention modules in each spatialtemporal DiT block. We use a uniform learning rate of $8 \times 10^{-5}$ during training. For Infini-Attention, we apply a learning rate of $3 \times 10^{-3}$ specifically to the global weight parameter to effectively balance global and local attention contributions while maintaining stable convergence. In VRAG, we set the weights as $[10.0, 10.0, 10.0, 3.0]$ across the global state dimentions ($[x, y, z, \text{yaw}]$) in the similarity function, to accommodate the wider range of yaw values. To differentiate retrieved historical frames from current context frames along the temporal dimension, we incorporate a temporal offset of 100 in the rotary position embeddings of temporal attentional for retrieved frames.

## C  Additional Experiments

### C.1  Analysis of Compounding Error Evaluation Metrics

Traditional metrics like SSIM, PSNR, and LPIPS measure pixel-level or feature-level differences between original and generated images. However, these metrics lose effectiveness when the generated video sample deviates significantly from the original video sample, especially for the compounding error evaluation, even if they falls in the same distribution and are visually reasonable. As shown in Figure 10, we normalize all metrics to a 0-1 scale where higher values indicate better generation quality (with SSIM score flipped). While all metrics perform well on the initial frame (index 0), assigning high scores to ground truth, their values begin to deteriorate after frame 100.

To address this limitation, we developed a discriminator-based evaluation metric. We train a discriminator using 1000 videos from the vanilla DF model (window size 20), with each video containing 1000 frames. This yielded a dataset of $10^6$ ground truth frames and $10^6$ generated frames as fake ones. We implemented the discriminator as a binary classifier using a lightweight architecture with 4 ResNet blocks. Too large discriminator architecture will lead to less meaningful discriminative signals. Each block contains two convolutional layers with batch normalization and activation functions. This design provides discriminative outputs while maintains computational efficiency.

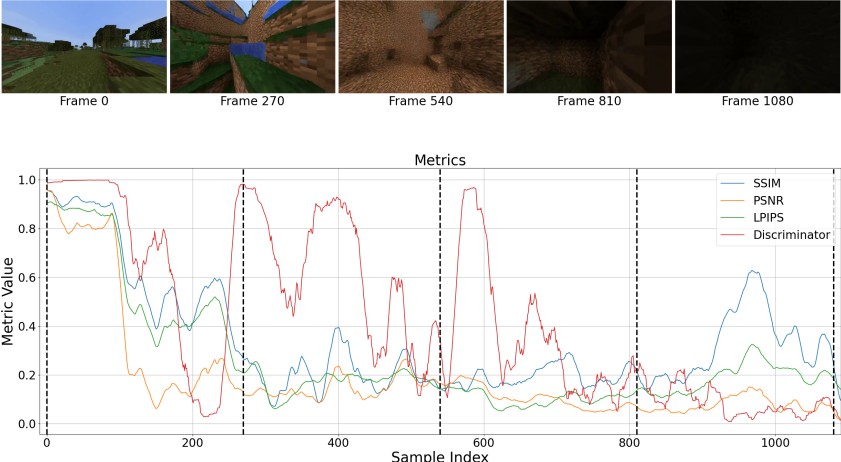

Figure 10: Comparison of SSIM, PSNR, LPIPS, and discriminator metrics. All metrics are normalized to the [0,1] range, where higher values indicate better performance for all scores. The discriminator score can accurately capture variations in generated image quality, while the other metrics are affected by distribution shift and fail to properly reflect compounding errors.

As shown in Fig. 10, the decrease of discriminator value faithfully reflects the distortions in generated images, while other metrics decline for two reasons: image quality degradation and distributional shift from the original video. This shift prevents traditional metrics from accurately assessing generation performance in terms of the compounding error. For instance, while the 270th frame shows significantly better generation quality than the 1080th frame, SSIM, PSNR, and LPIPS assign similar scores to both. This indicates that the distribution shift has become the dominant factor in lowering the metric scores, making these metrics unreliable for evaluating compounding error in long-range video generation.

Unlike traditional metrics, the discriminator's evaluation remains robust to distribution shifts since it doesn't depend on the original image, but rather depending only on the distortion of the generated images. This makes the discriminator score a more reliable metric for evaluating compounding errors in this case. However, the discriminator approach has several limitations. First, training requires sampling from a pre-trained diffusion model, which incurs computational overhead. Second, the training of the discriminator heavily depends on human judgment. We find that even a shallow ResNet architecture can effectively distinguish between ground truth and generated images. This suggests that an overly complex model might assign uniformly low scores to all generated content, making the discriminator metric less meaningful to look at. Finally, the discriminator shows limited generalization capability. When evaluating videos generated by new methods or datasets, the discriminator may be deceived into assigning inappropriately high scores. Therefore we do not report the discriminator score in the main paper, and advocate more investigation into faithful evaluation of compounding error in future work.

### C.2 Vanilla Long-context Extension vs. YaRN

To ensure a fair comparison, we evaluate YaRN against a baseline that directly extrapolates the vanilla model's window size from 20 during training to 40 at inference, to match the inference window length as YaRN in our experiments as Sec. 4. Evaluation of quantitative metrics LPIPS, SSIM and PSNR shown in Figures 11, 12, and 13 indicates that, YaRN maintains lower compounding error for long video generation (1100 frames). This demonstrates YaRN's effectiveness in extending the context window of diffusion video models to 40 frames after minimal fine-tuning. Vanilla extension of context length on DF models performs poorly due to out-of-distribution window size at inference.

While YaRN effectively extends the context window, its performance improvements are constrained by the inherent limitations of diffusion models in in-context learning. As demonstrated in Figure 14, the model exhibits difficulties in effectively leveraging long-range dependencies, leading to suboptimal spatialtemporal consistency against the ground truth. In addition, YaRN also requires greater

computational overheads during inference as it has a larger window size compared with other methods in our experiments in Sec. 4, making it less suitable for real-time gameplay applications.

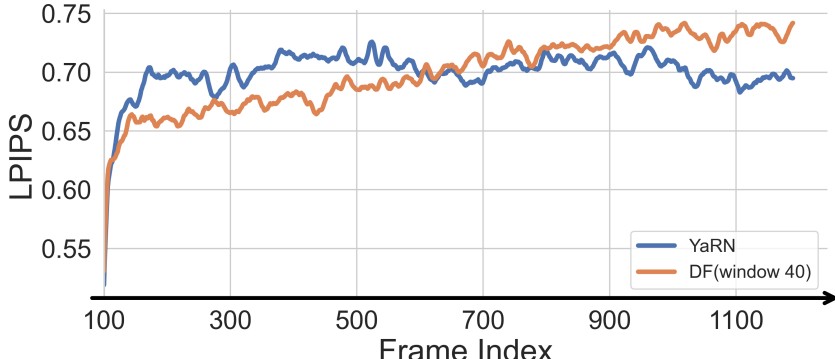

Figure 11: Comparison of vanilla long-context extension for DF model and YaRN with window length of 40 frames at inferences. Lower is better for LPIPS score.

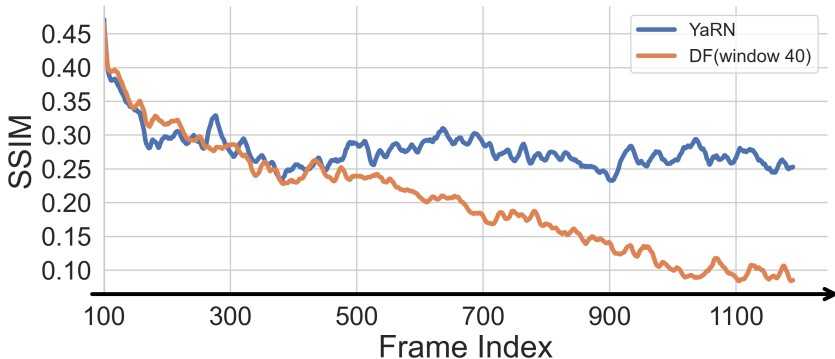

Figure 12: Comparison of vanilla long-context extension for DF model and YaRN with window length of 40 frames at inferences. Higher is better for SSIM score.

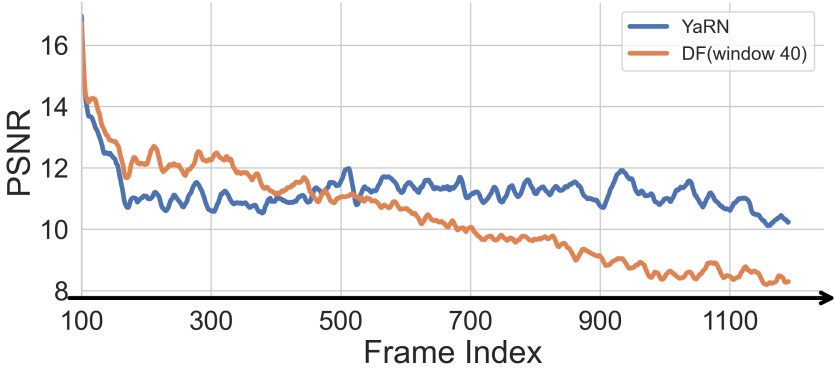

Figure 13: Comparison of vanilla long-context extension for DF model and YaRN with window length of 40 frames at inferences. Higher is better for PSNR score.

## C.3 More Discussions on Main Results

For the main results in Sec. 4.3 and Sec. 4.4, we provide more discussions here. The Infini-attention model faces significant training challenges due to its global attention mechanism. As evidenced in Figure 15, the model struggles to converge during training. For VRAG without memory component, we incorporated global state conditioning (specifically $[x, y, z, \text{yaw}]$) into the input. However,

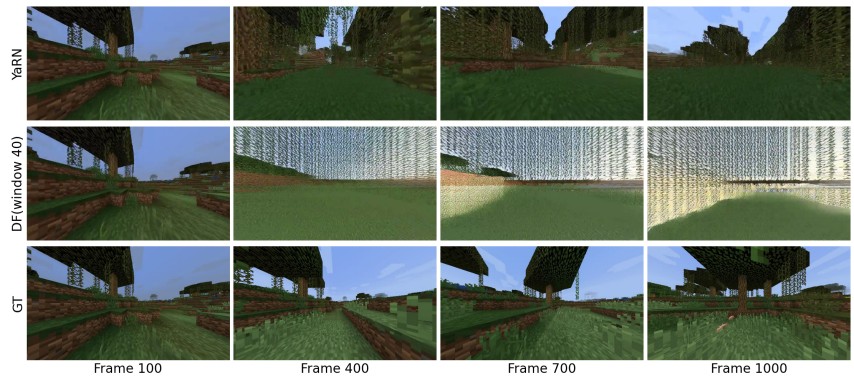

Figure 14: Visual comparison of vanilla long-context extension for DF model and YaRN. Both models are inferred with 40 frames window.

compared to the vanilla diffusion model, the training process becomes significantly more difficult. This may be due to the higher dimensionality and larger ranges of the spatial condition, whereas the action condition mostly consists of binary states ([0, 1]), making it harder for the model to learn and increasing perplexity.

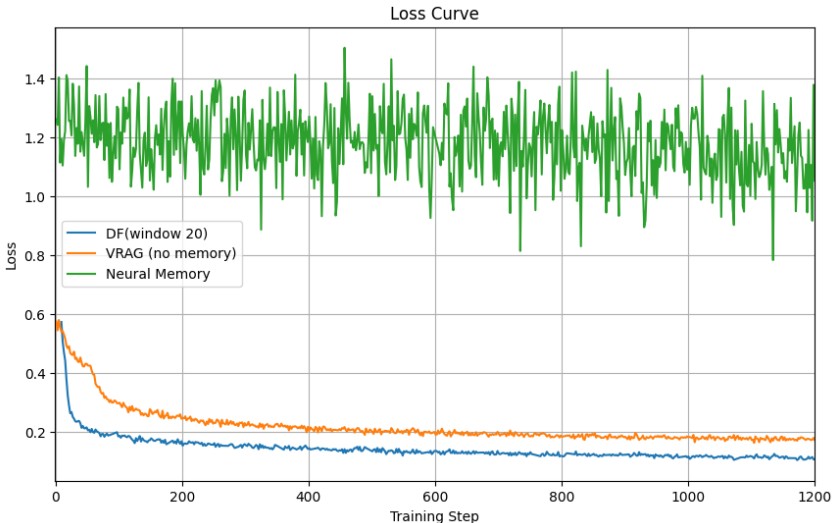

Figure 15: Training Loss Curves

## C.4 Predicted Global State

In the paper, our main experiments are conducted with the access to the ground-truth global state as conditions during training and inference. However, the practical usage may require the global state to be also predicted based on historical states and actions. To ablate this effect, we trained a pose (global state) prediction model that takes the current frame and action as inputs and outputs the predicted pose change. The next post can be derived by adding the predicted pose change to the current pose. Its architecture consists of only a few convolutional layers and fully connected layers, with a very small inference time overhead. At evaluation, we apply this trained predictor to predict the global state at next step, and generate videos based on the predicted global state. Following the same setting as in Sec. 4.3, the experimental results (for 300 frames prediction) are summarized in Table 6.

| Method | SSIM↑ | PSNR↑ | LPIPS↓ |
|---|---|---|---|
| DF20 | 0.466 | 16.643 | 0.538 |
| VRAG (predicted pose) | 0.500 | **17.116** | **0.506** |
| VRAG | **0.506** | 17.097 | **0.506** |

Table 6: Ablation study of replacing the ground-truth global state with predicted ones by a trained pose predictor.

As shown in the table, the evaluation results are nearly identical with or without the pose prediction, since the pose prediction is a relatively simple task compared with the video generation. This proves the feasibility of using the predicted global state without significant video performance degradation.

### C.5 Memory and Time Overhead

We also compare the memory usage and inference time of VRAG against several baselines: diffusion forcing with 10 and 20 context frames, and YaRN with 40 context frames.

| Method | DF10 | DF20 | YaRN | VRAG |
|---|---|---|---|---|
| Context length (Frame) | 10 | 20 | 40 | 20 |
| Memory usage (MB) | 4420 | 4448 | 4543 | 4452 |
| Inference Time (min) | 9 | 12 | 23 | 12 |

Table 7: Memory usage and inference-time of different methods

As demonstrated in the Table 7, VRAG's GPU memory usage and inference time overhead are nearly identical to DF20. The inference time is derived for autoregressive generation over 600 frames. Meanwhile, the computation for the retrieval operations can be entirely performed on the CPU, with its memory footprint being only num_frame × action_dim × 4 Bytes = 9.4 KB in our experiments, which is almost negligible.

In summary, VRAG incurs almost no additional inference-time overhead compared to standard diffusion forcing. The memory and computational cost introduced by the retrieval mechanism are negligible, as it only involves similarity calculations between a set of vectors.

## D More Results

| Method | SSIM ↑ | PSNR ↑ | LPIPS ↓ |
|---|---|---|---|
| VRAG | **0.349** | **12.039** | **0.654** |
| VRAG (no training) | 0.218 | 11.588 | 0.712 |
| VRAG (no memory) | 0.205 | 11.367 | 0.746 |

Table 8: Ablation study of VRAG components for compounding error on long video generation. We compare the full model with variants that remove either the memory component (additional global state conditioning only) or training component (in-context learning only).

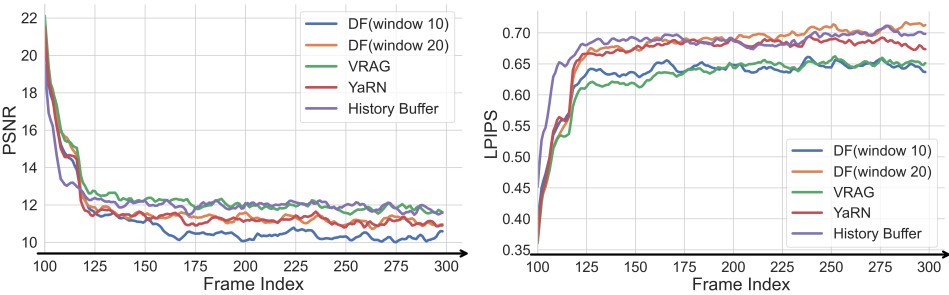

Figure 16: World coherence evaluation on all methods for PSNR (left) and LPIPS (right).

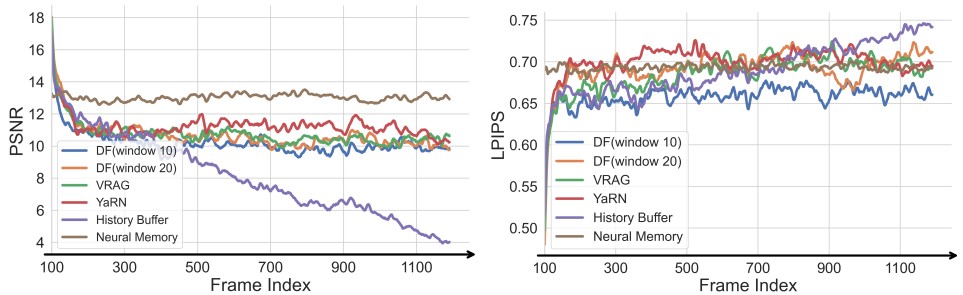

Figure 17: Compounding error evaluation on all methods for PSNR (left) and LPIPS (right).

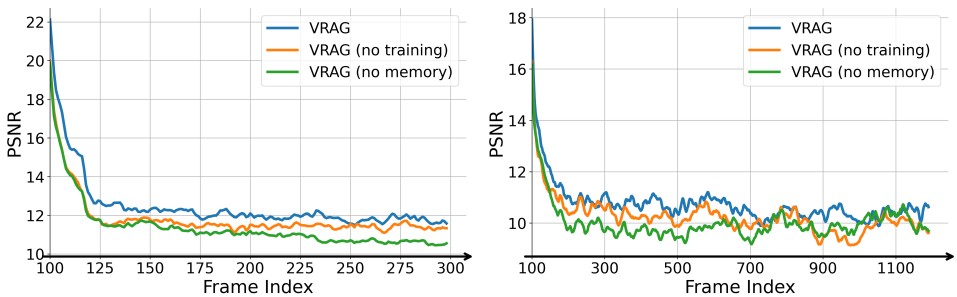

Figure 18: Ablation study of VRAG components for world coherence (left) and compounding error (right), with PSNR metric. We compare the full model with variants that remove either the memory component (additional global state conditioning only) or training component (in-context learning only).

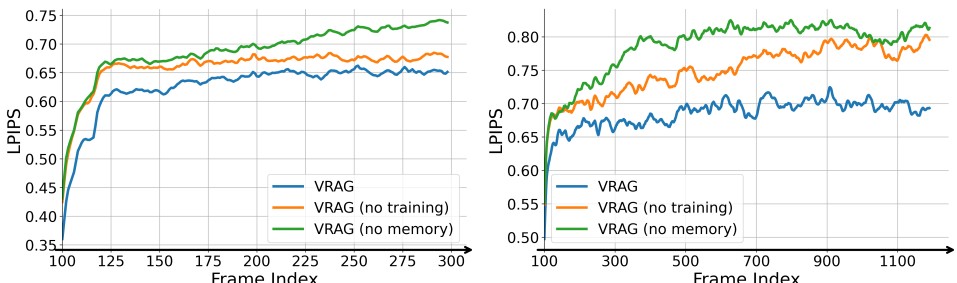

Figure 19: Ablation study of VRAG components for world coherence (left) and compounding error (right), with LPIPS metric. We compare the full model with variants that remove either the memory component (additional global state conditioning only) or training component (in-context learning only).

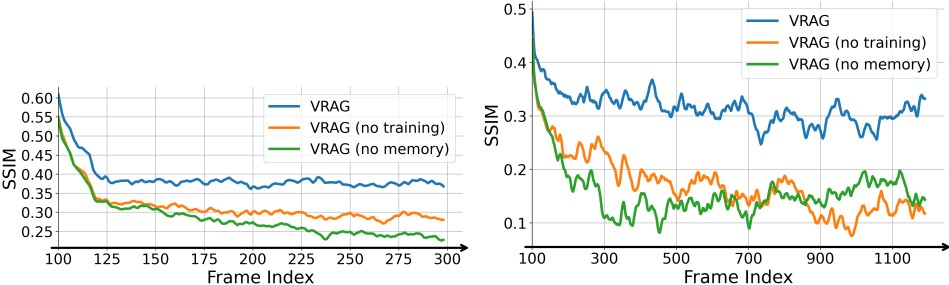

Figure 20: Ablation study of VRAG components for world coherence (left) and compounding error (right), with SSIM metric. We compare the full model with variants that remove either the memory component (additional global state conditioning only) or training component (in-context learning only).

