# OpenReview forum: "Learning World Models for Interactive Video Generation"
_NeurIPS.cc/2025/Conference — NeurIPS 2025 poster_

### Official Review · Reviewer_4P6H · 2025-06-11

**Clarity:** 3
**Significance:** 2
**Originality:** 2
**Rating:** 4
**Confidence:** 3

**Summary:**

This work introduces Video Retrieval Augmented Generation (VRAG), a framework for long-horizon, action-conditioned video generation with improved spatio-temporal consistency. The authors identify two key challenges in autoregressive video world models: compounding errors over time and insufficient memory mechanisms for maintaining long-term coherence. VRAG mitigates these by (1) conditioning on explicit global state (position and pose), and (2) retrieving and integrating past frames through a modified memory buffer during both training and inference. Through extensive experiments in Minecraft, the authors demonstrate that standard long-context techniques from language models (e.g. RoPE extensions, neural memory) are ineffective in the video domain, while VRAG substantially improves video quality and long-term consistency.

**Questions:**

* Would VRAG work in domains without access to structured global state (raw RGB-only settings)?

* Have you evaluated whether global state prediction (for example from vision) can substitute the oracle state used in your setup?

* What is the inference-time overhead introduced by retrieval, especially during long-horizon rollouts?

**Ethical Concerns:**

["NO or VERY MINOR ethics concerns only"]

**Final Justification:**

I would like to thank the authors for responding to my questions, which have all been well addressed. After reading the other reviews, that acknowledged similar strengths to what I've highlighted, and no particular strong weakness that have not been left unanswered in the rebuttal, I've decided to keep my initial rating.

**Limitations:**

yes

**Paper Formatting Concerns:**

/

**Quality:**

3

**Strengths And Weaknesses:**

### __Strengths__

1. The paper tackles two fundamental issues: compounding error and long term memory, through an interactive and long video generation setting.

2. VRAG outperforms strong baselines (Diffusion Forcing, Neural Memory) across world coherence and compounding error evaluations.

3. The authors convincingly show that in-context learning techniques from LLMs transfer poorly to video generation.

### __Weaknesses__

1. The effectiveness of VRAG depends on access to explicit global states (like coordinates and orientation), which may not be available or easily inferred in real-world settings.

2. There is no qualitative video to support the claims of the paper, apart from the image figures. It would be much more compelling to have a side-by-side comparison of the LLM techniques and the proposed VRAG in a long horizon rollout.

3. The method is only evaluated on Minecraft gameplay, a synthetic and stylised environment. It’s unclear whether the findings would hold in real-world video (robotics, driving).

---

> ### Author Rebuttal · Authors · 2025-07-31
>
> > W1: The effectiveness of VRAG depends on access to explicit global states (like coordinates and orientation), which may not be available or easily inferred in real-world settings.
> >
>
> **Reply to W1:** We sincerely appreciate your insightful concern regarding the dependency on explicit global states. It is worth noting that VRAG can still be effectively employed even when such states are not directly available. For instance, retrieval can be performed based on image similarity to access relevant historical information without relying on explicit coordinates. Additionally, as mentioned in our response to Q2, a lightweight pose predictor can be trained to infer the necessary global state information. Furthermore, with the rapid advancements in SLAM algorithms, many RGB-only SLAM methods have proven highly effective in accurately estimating global states. In fact, as discussed in our response to W3, the RealEstate10K dataset itself leverages SLAM-derived camera trajectories to obtain global state information, demonstrating the feasibility of this approach in real-world settings.
>
> > W2: There is no qualitative video to support the claims of the paper, apart from the image figures. It would be much more compelling to have a side-by-side comparison of the LLM techniques and the proposed VRAG in a long horizon rollout.
> >
>
> **Reply to W2:** We have prepared some video-based qualitative results for visualization. However, we regret to notice that due to NIPS's newly implemented policy this year, we are unable to provide any external links or any images/videos. We plan to create a project webpage in the future where these video demonstrations will be made available. If interested, please follow our work for updates.
>
> > W3: The method is only evaluated on Minecraft gameplay, a synthetic and stylised environment. It’s unclear whether the findings would hold in real-world video (robotics, driving).
> >
>
> **Reply to W3:** We fully understand your concerns and appreciate your valuable suggestions. However, on one hand, driving datasets are extremely scarce, and most of them are internal data that is not open-source. On the other hand, most robotic manipulation scenarios are relatively monotonous with less dynamic backgrounds, and the phenomenon of forgetting occurs less frequently. Therefore, we follow the experimental setup of **Diffusion Forcing Transformer (DFoT)** [1]. We trained our diffusion model on the RealEstate10K dataset and used the pretrained model from [1] as the baseline. After fine-tuning for just 2 epochs (10% of the original training steps), our method significantly outperforms the baseline in terms of memorization capability. The results (summarized in the table below) demonstrate that our approach generalizes effectively beyond the Minecraft domain.
>
> | Method\Metric | SSIM↑ | PSNR↑ | LPIPS↓ | FVD↓ |
> | --- | --- | --- | --- | --- |
> | DFoT | 0.4436 | 13.03 | 0.4469 | 337.5 |
> | VRAG | **0.9116** | **32.21** | **0.1146** | **221** |
>
> > Q1: Would VRAG work in domains without access to structured global state (raw RGB-only settings)?
> >
>
> **Reply to Q1:** As addressed in our response to W1, we believe VRAG remains applicable even in scenarios where explicit global states are unavailable.
>
> > Q2: Have you evaluated whether global state prediction (for example from vision) can substitute the oracle state used in your setup?
> >
>
> **Reply to Q2:** We trained a pose predictor model that takes actions and images as inputs. Its architecture consists of only a few convolutional layers and fully connected layers, resulting in a very small inference time overhead. We used the trained predictor to make predictions on inference data and generated videos using the predicted global state. The experimental results (for 300 frames prediction) are as follows:
>
> | Method\Metric | SSIM↑ | PSNR↑ | LPIPS↓ |
> | --- | --- | --- | --- |
> | DF20 | 0.466 | 16.643 | 0.538 |
> | VRAG (predicted pose) | 0.500 | **17.116** | 0.506 |
> | VRAG | **0.506** | 17.097 | **0.506** |
>
> As we can see, the evaluation results are nearly identical. This proves the feasibility of using the predicted global state. We will add this experiments in our paper for clarity.
>
> > Q3: What is the inference-time overhead introduced by retrieval, especially during long-horizon rollouts?
> >
>
> **Reply to Q3:** During single-step model prediction, VRAG incurs almost no additional inference-time overhead compared to standard diffusion forcing. The computational cost introduced by the retrieval mechanism is negligible, as it only involves similarity calculations between a set of vectors.
>
> | Method | DF20 | VRAG |
> | --- | --- | --- |
> | Memory usage (MB) | 4448 | 4452 |
> | Inference Time (min) | 12 | 12 |
>
> As demonstrated in the table, VRAG's GPU memory usage and inference time overhead are nearly identical to DF20. The inference time is derived for autoregressive generation over 600 frames. Meanwhile, the computation for the retrieval part can be entirely performed on the CPU, with its memory footprint being only num_frame × action_dim × 4 Bytes = 9.4 KB, which is almost negligible.
>
> **Reference**
>
> [1] History-Guided Video Diffusion, Arxiv 2025

---

> > ### Comment · Reviewer_4P6H · 2025-08-05
> >
> > Thank you very much for your detailed and precise responses regarding explicit global state, the inference time of the model, and the applicability of the proposed method to non-synthetic datasets. All my concerns have been addressed.

---

### Official Review · Reviewer_M9fd · 2025-06-30

**Clarity:** 2
**Significance:** 2
**Originality:** 2
**Rating:** 4
**Confidence:** 4

**Summary:**

This paper proposes several methods to address the challenges of compounding errors and insufficient memory in autoregressive, action-conditioned video generation. These methods include action and state conditioning, history state retrieval, and so on. Experiments are conducted on the Minecraft datasets, and the performance is evaluated based on several video-quality metrics.

**Questions:**

- How's the performance of the proposed retrieval method compared to FramePack [1]? This baseline should be included in the next version of this paper.
- Can the authors provide more qualitative demos for assessing the performance, such as the generated videos?
- Is the model trained from scratch or adapted from a pre-trained model? If this model is trained from scratch, why don't the authors choose a pre-trained model for fine-tuning?

[1] Packing Input Frame Context in Next-Frame Prediction Models for Video Generation

**Ethical Concerns:**

["NO or VERY MINOR ethics concerns only"]

**Final Justification:**

The authors addressed some of my concerns in the discussion phase, thus I increased the rating accordingly. However, the main contributions and insights provided are still not clear in this version. I believe this paper should incorporate the additional experiments, explanations, and discussions in the next version to reach the acceptance threshold.

**Limitations:**

Limitations have been discussed in the paper

**Quality:**

2

**Strengths And Weaknesses:**

## Strengths
- This paper identifies two key challenges in autoregressive video generation: compounding errors and insufficient memory. Some analyses are given for addressing these challenges.
- This paper incorporates methods from LLMs to investigate their performance in the context of video generation, which is a promising direction.
- The proposed method is sound. I like the idea of history retrieval, which could enhance the in-context learning ability for video generative models.
## Weaknesses
- Writing in this paper needs to be improved. The paper should highlight the insights and contributions. However, there are many paragraphs describing different methods, training details, baselines, and so on.
- Experimental results are not sufficient to validate the effects of the proposed method. I noticed that the background and diversity of the Minecraft videos are limited. Therefore, **I suspect the trained model can be easily overfitted and can be difficult to generalize to other domains**. Moreover, the paper only presents the empirical results for in-domain performance, where the metrics are calculated by comparison with the ground truth videos.
- In line 160, this paper claims that global state conditioning is significant for the final performance. However, for many realistic cases where simulators are not available, this kind of information is not accessible.
- Miss references. The method of action-conditioned block has been discussed by previous works [1,2], and the idea of leveraging video models for world simulation has been proposed by a recent work [1]. This paper should add a discussion section to distinguish the differences and advantages compared to existing works.

[1] Pre-Trained Video Generative Models as World Simulators, Arxiv 2025

[2] IRASim: Learning Interactive Real-Robot Action Simulators, Arxiv 2024

---

> ### Author Rebuttal · Authors · 2025-07-31
>
> > W1: Writing in this paper needs to be improved...
>
> **Reply to W1:** We appreciate the reviewer’s feedback on further improving the writing. We will modify accordingly to highlight more on our contributions and leave more details of different methods and implementation in supplementary materials.
>
> > W2: Experimental results are not sufficient to validate the effects of the proposed method...
>
> **Reply to W2:**  We appreciate the reviewer’s concerns regarding the diversity of Minecraft videos and the generalizability of our method. To address these points, we have conducted additional experiments in a real-world setting. Specifically, following the experimental setup of **Diffusion Forcing Transformer (DFoT)** [4], we trained our diffusion model on the RealEstate10K dataset and used the pretrained model from [4] as the baseline. After fine-tuning for just 2 epochs (10% of the original training steps), our method significantly outperforms the baseline in terms of memorization capability. The results (summarized in the table below) demonstrate that our approach generalizes effectively beyond the Minecraft domain.
>
> |Method\Metric|SSIM↑|PSNR↑|LPIPS↓|FVD↓|
> |-|-|-|-|-|
> |DFoT|0.4436|13.03|0.4469|337.5|
> |VRAG|**0.9116**|**32.21**|**0.1146**|**221**|
>
> Regarding the concern about overfitting, we believe there may be a misunderstanding of our methods and evaluation protocal. Our test set is entirely distinct from the training set, and the context frames are different from those to be predicted for both training and test stages. Importantly, our goal is to let the model develop **in-context learning** capabilities at inference time—retaining and recalling predicted historical information during inference, instead of reusing ground truth frames. For example, if the model sees a tree in its historical context, we expect it to generate the same tree from the same viewpoint later, relying on retrieval and in-context learning rather than just memorization of training data. Overfitting on the training set would not provide this in-context learning capability.
>
> Finally, we acknowledge the reviewer’s observation that our evaluation metrics primarily compare generated videos to ground truth. While this is a valid consideration, we want to emphasize three points: (1) We did not claim to build a foundational world model across domains due to the limited resources, but focus on validating the effectiveness of VRAG in single domain (Minecraft) setting, therefore the evaluation is indeed in-domain performance. (2) One of our major contributions is improving the model’s memorization capability, therefore frame-wise measure against the ground truth video is a natural and valid metric for fidelity. (3) Specifically, for long-term video consistency, there is no commonly applied evaluation metric, and most existing work relies on ground truth comparisons, with metrics reported in our paper. To further strengthen our evaluation, we have incorporated additional metrics from **VBench** [5] to assess temporal quality and video frame quality. Since our work does not focus on text-to-video generation, we did not employ the Video-Condition Consistency metrics from VBench, but instead focused on the Video Quality metrics. The evaluation results on VBench (higher is better) are shown in the following table:
>
> |Method\Metric|Background Consistency|Temporal Flickering|Motion Smoothness|Aesthetic Quality|Imaging Quality|
> |-|-|-|-|-|-|
> |DF20|0.9668|0.9485|0.9582|0.5272|0.6058|
> |YaRN|**0.9686**|0.9401|0.9523 |0.5252|0.6323|
> |Historical Buffer|0.9664|0.9475|0.9579|0.5167|0.6253|
> |**VRAG**|**0.9686**|**0.9511**|**0.9603**| **0.5295**|**0.6444**|
>
> As demonstrated in the results, our method outperforms all other baselines across all metrics in both temporal quality and video frame quality. The Neural Memory baseline has Aesthetic Quality 0.343 and Imaging Quality score 0.3597 respectively, which are significantly lower than other baselines, therefore not listed here. Our VRAG shows better temporal consistency compared with all baseline methods, and the high video frame quality indicates the results are not over-smoothed.
>
> We hope these clarifications and additional experiments alleviate the reviewer’s concerns.
>
> > W3: In line 160, this paper claims that global state conditioning is significant...
>
> **Reply to W3:** We sincerely appreciate the reviewer's concern. We would like to clarify that while explicitly modeling global state is crucial for accurately memorizing and retrieving generated videos, our VRAG framework does not solely rely on global state to be effective. The system can still retrieve relevant historical information based on frame similarity alone, which no longer requires the global state in retrieval process.
>
> Moreover, there are various methods to extract global state information from RGB videos, such as SLAM algorithms. As mentioned in W2's response, the RealEstate10K dataset actually utilizes SLAM to collect global state information. Same technique can be applied on other datasets as well. Additionally, we have trained a lightweight pose prediction model, to replace the ground-truth global state during model inference, and the experimental results (see table) demonstrate that the predictor can achieve comparable performance as our current methods for 300 frames prediction. We will provide this additional experiments in our paper to alleviate the concern.
>
> | Method\Metric|SSIM↑|PSNR↑|LPIPS↓|
> |-|-|-|-|
> | DF20 |0.466|16.643| 0.538 |
> | VRAG (predicted pose)|0.500|**17.116**|**0.506**|
> | VRAG|**0.506**|17.097|**0.506**|
>
> > W4: Miss references...
>
> **Reply to W4:** We appreciate the reviewer for mentioning the missing references and will add discussions in our paper. The two papers focus on robotic manipulation video prediction, with less dynamic backgrounds. Although they also work on action-conditioned video prediction like our baseline methods (DF20), they do not try to address the memory issue in long-term world modeling. Our work distinguish from theirs in both dataset choices and techniques.
>
> > Q1: How's the performance of the proposed retrieval method compared to FramePack? This baseline should be included in the next version of this paper.
>
> **Reply to Q1:** We sincerely appreciate the reviewer's valuable suggestion. We have additionally conducted experiments using Frame Pack [3] as a baseline with same number of context frames and training steps. The comparative results shown in the following table:
>
> |Method\Metric|SSIM↑|PSNR↑|LPIPS↓|
> |-|-|-|-|
> |Frame Pack|0.421|16.372|0.574|
> |DF20| 0.466|16.643|0.538|
> |VRAG|**0.506**|**17.097**|**0.506**|
>
> Our VRAG clearly outperforms this baseline across metrics under the same setting. We would also like to emphasize that, at the time of our original submission, the Frame Pack paper is a concurrent work. Also, the key techniques of Frame Pack and ours are orthogonal in nature - while Frame Pack primarily focuses on improving computational efficiency and reducing overhead, our method can actually address the memory issue more effectively. Our proposed video retrieval augmentation can be complementary to the Frame Pack technique for better performances as a future work.
>
> As noted in the Frame Pack paper, their frame compression relies on importance sampling that typically prioritizes more recent frames, which is straightforward and not optimal. When combined with our approach, the similarity function from our VRAG framework could be a better alternative, which would significantly enhance its memory capabilities with recall of the most relevant information in historical frames. This potential synergy demonstrates how our methods can work together rather than a direct competitor.
>
> > Q2: Can the authors provide more qualitative demos for assessing the performance, such as the generated videos?
>
> **Reply to Q2:** We have prepared some video-based qualitative results for visualization. However, we regret to notice that due to NIPS's newly implemented policy this year, we are unable to provide any external links or any images/videos. We plan to create a project webpage in the future where these video demonstrations will be made available. If interested, please follow our work for updates.
>
> > Q3: Is the model trained from scratch or adapted from a pre-trained model? If this model is trained from scratch, why don't the authors choose a pre-trained model for fine-tuning?
>
> **Reply to Q3:** All models in our study are trained from scratch. We sincerely appreciate the reviewer's valuable suggestion regarding the use of pre-trained models. Indeed, fine-tuning from pre-trained models represents a promising direction that could potentially enhance both efficiency and performance. As shown in our additional experiments with DFoT fine-tuning (please refer to the results in W2's response), we have explored this approach.
>
> There are two primary reasons for our original decision not to use pre-trained models. First, there are currently limited open-source models employing the diffusion forcing paradigm. While fine-tuning from full sequence denoising models is a viable strategy, it requires further investigation. Additionally, some of our baseline methods cannot be directly adapted from pre-trained models through fine-tuning due to their unique architectural designs, such as YaRN and Neural Memory. Second, the Minecraft environment in our study features relatively simple settings, where training from scratch can achieve satisfactory results, as demonstrated by our experimental outcomes.
>
> **Reference**
>
> [1] Pre-Trained Video Generative Models as World Simulators, Arxiv 2025
>
> [2] IRASim: Learning Interactive Real-Robot Action Simulators, Arxiv 2024
>
> [3] Packing Input Frame Context in Next-Frame Prediction Models for Video Generation, Arxiv 2025
>
> [4] History-Guided Video Diffusion, Arxiv 2025
>
> [5] VBench: Comprehensive Benchmark Suite for Video Generative Models, CVPR 2024

---

> > ### Comment · Reviewer_M9fd · 2025-08-05
> >
> > Thanks for your additional experiments to address my concerns. I will increase my score accordingly.

---

### Official Review · Reviewer_UWPa · 2025-06-30

**Clarity:** 3
**Significance:** 2
**Originality:** 2
**Rating:** 4
**Confidence:** 3

**Summary:**

The paper observes that existing video generation models, when used as interactive world models, exhibit pronounced spatiotemporal inconsistencies. It traces these issues to two shortcomings of autoregressive generators: compounding prediction errors and an inadequate memory mechanism. To overcome these limitations, the authors propose Video Retrieval-Augmented Generation (VRAG), which enriches the video world model with global state conditioning and memory retrieval mechanisms, thereby achieving long-term consistency.

**Questions:**

(1)	All reported metrics—SSIM, PSNR, and LPIPS—are frame-level measures and therefore do not adequately capture improvements in long-term consistency. It would be better to provide a justification for why these metrics are suitable for evaluating long-term consistency, or introduce and employ improved metrics that better capture long-term consistency.

(2) The experiments are conducted on only a single dataset, which may restrict the generalizability of the results. To strengthen the claims, the authors may consider validate their approach on additional datasets.

(3) The range of comparison methods is too limited—experiments compare only against a few variants of the baseline (the vanilla model in Sec. 3.2).  For a more comprehensive evaluation, it would be better to include comparisons with a broader range of state-of-the-art methods.

**Ethical Concerns:**

["NO or VERY MINOR ethics concerns only"]

**Final Justification:**

All my concerns about metrics and comprison on more datasets are well answered.

**Limitations:**

yes

**Quality:**

2

**Strengths And Weaknesses:**

Strengths:
(1) The authors seamlessly transfer techniques from the LLM to the autoregressive-generation setting and refine them so they operate effectively in that regime.
(2) The paper conducts a thorough ablation study, demonstrating the effectiveness of the proposed method.

Weaknesses:
(1) All reported metrics—SSIM, PSNR, and LPIPS—are frame-level measures and therefore do not adequately capture improvements in long-term consistency. Maybe an analysis between the metrics and
(2) Experiments are restricted to a single dataset, which limits the generalizability of the findings.
(3) The range of comparison methods is too limited—experiments compare only against a few variants of the baseline (the vanilla model in Sec. 3.2).

---

> ### Author Rebuttal · Authors · 2025-07-31
>
> > Q1: All reported metrics—SSIM, PSNR, and LPIPS—are frame-level measures and therefore do not adequately capture improvements in long-term consistency. It would be better to provide a justification for why these metrics are suitable for evaluating long-term consistency, or introduce and employ improved metrics that better capture long-term consistency.
> >
>
> **Reply to Q1:** We sincerely appreciate the reviewer's valuable concerns and suggestions regarding the evaluation metrics. These metrics are commonly applied in existing research of video world modeling [4,5,6,7]. Introducing improved metrics for long-term consistency is indeed important but it could be an independent research topic. The frame level distance against ground truth indicates the fidelity of prediction regardless of the video length. However, the temporal consistency is indeed lack in our measure. To address this concern, we have conducted additional evaluations using metrics from **VBench** [1] to assess our generated videos, including temporal quality metrics like Background Consistency, Temporal Flickering and Motion Smoothness. Since our work does not focus on text-to-video generation, we did not employ the Video-Condition Consistency metrics from VBench, but instead focused on the Video Quality metrics. The evaluation results on VBench (higher is better) are shown in the following table:
>
> | Method\Metric | Background Consistency | Temporal Flickering | Motion Smoothness | **Aesthetic Quality** | **Imaging Quality** |
> | --- | --- | --- | --- | --- | --- |
> | DF20 | 0.9668 | 0.9485 | 0.9582 | 0.5272 | 0.6058 |
> | YaRN | **0.9686** | 0.9401 | 0.9523 | 0.5252 | 0.6323 |
> | Historical Buffer | 0.9664 | 0.9475 | 0.9579 | 0.5167 | 0.6253 |
> | **VRAG** | **0.9686** | **0.9511** | **0.9603** | **0.5295** | **0.6444** |
>
> As demonstrated in the results, our method outperforms all other baselines across all metrics in both temporal quality and video frame quality. The Neural Memory baseline has Aesthetic Quality 0.343 and Imaging Quality score 0.3597 respectively, which are significantly lower than other baselines, therefore not listed here. Our VRAG shows better temporal consistency compared with all baseline methods, and the high video frame quality indicates the results are not over-smoothed.
>
> > Q2: The experiments are conducted on only a single dataset, which may restrict the generalizability of the results. To strengthen the claims, the authors may consider validate their approach on additional datasets.
> >
>
> **Reply to Q2:** We sincerely appreciate your concern regarding the generalizability of our approach. To address this, we have conducted additional experiments in a real-world setting. Specifically, following the experimental setup of **Diffusion Forcing Transformer (DFoT)** [2], we trained our diffusion model on the RealEstate10K dataset and used the pretrained model provided in [2] as the baseline.
>
> After fine-tuning for just 2 epochs (10% of the original training steps), our method significantly outperforms the baseline in terms of memorization capability. The experimental results are summarized in the table below:
>
> | Method\Metric | SSIM↑ | PSNR↑ | LPIPS↓ | FVD↓ |
> | --- | --- | --- | --- | --- |
> | DFoT | 0.4436 | 13.03 | 0.4469 | 337.5 |
> | VRAG | **0.9116** | **32.21** | **0.1146** | **221** |
>
> We believe these results effectively demonstrate the generalizability of our approach beyond Minecraft.
>
> > Q3: The range of comparison methods is too limited—experiments compare only against a few variants of the baseline (the vanilla model in Sec. 3.2). For a more comprehensive evaluation, it would be better to include comparisons with a broader range of state-of-the-art methods.
> >
>
> **Reply to Q3:** We sincerely appreciate the reviewer's concern regarding the comprehensiveness of our baseline methods. To address this, we have expanded our experiments by incorporating two additional baselines to strengthen our paper. Specifically, we have supplemented experiments on RealEstate and adopted the Diffusion Forcing Transformer [2] as one of our baselines (please refer to our response to Q2 for details). For the experiments in Minecraft, following Reviewer M9fd's suggestion, we have included Frame Pack [3] as another baseline to further demonstrate the superiority of our approach. The detailed experimental results are presented in the table below, and our method is superior to the Frame Pack baseline.
>
> | Method\Metric | SSIM↑ | PSNR↑ | LPIPS↓ |
> | --- | --- | --- | --- |
> | Frame Pack | 0.421 | 16.372 | 0.574 |
> | DF20 | 0.466 | 16.643 | 0.538 |
> | VRAG | **0.506** | **17.097** | **0.506** |
>
> **Reference**
>
> [1] VBench: Comprehensive Benchmark Suite for Video Generative Models, CVPR 2024
>
> [2] History-Guided Video Diffusion, Arxiv 2025
>
> [3] Packing Input Frame Context in Next-Frame Prediction Models for Video Generation, Arxiv 2025
>
> [4] Stable Video Diffusion: Scaling Latent Video Diffusion Models to Large Datasets, Arxiv 2023
>
> [5] CogVideoX: Text-to-Video Diffusion Models with An Expert Transformer, ICLR 2025
>
> [6] SlowFast-VGen: Slow-Fast Learning for Action-Driven Long Video Generation, Arxiv 2024
>
> [7] GEN3C: 3D-Informed World-Consistent Video Generation with Precise Camera Control, Arxiv 2025

---

> > ### Comment · Reviewer_UWPa · 2025-08-05
> > **Official Comments by Reviewer UWPa**
> >
> > All my conerns are well explained. And It is suggested to include more methods for comparision on RealEstate10K.

---

### Official Review · Reviewer_udTT · 2025-07-02

**Clarity:** 3
**Significance:** 3
**Originality:** 2
**Rating:** 4
**Confidence:** 4

**Summary:**

This paper focuses on world models (action conditioned interactive video generation), training models in the minecraft domain. The authors propose VRAG, comprising of two approaches to improve memory - a RAG based system and some game state conditioning. The model is based off external baselines like diffusion forcing. The ideas are neat, and seem to help in this setting, although questions remain about their general applicability given the extremely limited dataset used for the results.

**Questions:**

Do you have any fair comparisons w.r.t memory usage for your method vs. baselines? I assume yours uses more, for the same model size, so it is not a fair comparison.

What if you scale up the baseline?

**Ethical Concerns:**

["NO or VERY MINOR ethics concerns only"]

**Limitations:**

Compute/dataset size is so small that this might not be relevant for the frontier of the field.

**Quality:**

3

**Strengths And Weaknesses:**

Strengths
- The paper is in a relevant area, as world models and interactive video generation are becoming increasingly important.
- The model ideas seem novel, borrowing concepts from LLMs (e.g. RAG) and showing it can be effectively applied to this new setting.
- Experiments show clear ablations which means actionable insights can be taken and applied to the more general domain. Further, this work may help folks train better world models for single gaming environments, which is a current popular paradigm.

Weaknesses
- The dataset size is trivially small, which makes all the analysis potentially limited to the specific small scale setting. The methods proposed might be very effective for memorizing the data, rather than providing generalization benefits at scale. To be concrete, the dataset is 17 hours of minecraft, vs. Genie 1 which used 30,000 hours of internet videos from hundreds of games. Genie 1 is also 18 months old and not SoTA.
- It is definitely a negative that this work focuses entirely on Minecraft. We cannot be sure if ideas generalize to a truly “foundation” world model if it is only trained on a single game.
- The citations are totally off in this paper, which makes the positioning very strange. In the opening statement, the authors say “Foundational world models capable of simulating future outcomes based on different actions 21 are crucial for effective planning and decision making [1, 2, 3].” and then cite [1] a NeurIPS 2015 paper which is not a foundation world model, [2] David Ha World Models paper, which is iconic, but also not a foundation world model, and then [3] Dreamer, which is a single environment, not foundation world model. A foundation world model is one where it can be prompted to generate new worlds, which none of these can. The only foundation world models produced so far are Genie1, Genie2 and Cosmos, as far as I am aware.
- The authors incorrectly reference the Genie 2 model “One line of research addressing this involves predicting the underlying 3D world structure like Genie2”. This is incorrect, Genie 2 was a latent diffusion model, with no inductive bias. If anything, it is very close related work.

---

> ### Author Rebuttal · Authors · 2025-07-31
>
> > W1: The dataset size is trivially small, which makes all the analysis potentially limited to the specific small scale setting. The methods proposed might be very effective for memorizing the data, rather than providing generalization benefits at scale. To be concrete, the dataset is 17 hours of minecraft, vs. Genie 1 which used 30,000 hours of internet videos from hundreds of games. Genie 1 is also 18 months old and not SoTA.
> >
>
> **Reply to W1:** We appreciate the reviewer’s feedback regarding dataset size and scalability. As an academic research group, our computational resources are inherently limited compared to industry-scale projects like Genie, which leverages Google’s infrastructure for 30K hours of training data. Our goal is not to achieve state-of-the-art generalization across diverse domains but to validate our method’s efficacy specifically for compounding errors and memory issues in modeling Minecraft—a well-defined, structured environment where 17 hours of gameplay (though modest) suffices to demonstrate meaningful world modeling capabilities. The results show our model captures key dynamics without overfitting, suggesting the approach is viable for targeted domains even with smaller-scale data. We believe our method can show more significant performance gain at a larger scale. Scaling to internet-level datasets remains valuable future work, but our focus here is on foundational insights for domain-specific applications.
>
> > W2: It is definitely a negative that this work focuses entirely on Minecraft. We cannot be sure if ideas generalize to a truly “foundation” world model if it is only trained on a single game.
> >
>
> **Reply to W2:** We sincerely appreciate your concern regarding the generalizability of our approach. To address this, we have conducted additional experiments in a real-world setting. Specifically, following the experimental setup of **Diffusion Forcing Transformer (DFoT)** [1], we trained our diffusion models on the RealEstate10K dataset and used the pretrained model provided in [1] as the baseline. After fine-tuning for just 2 epochs (10% of the original training steps), our method significantly outperforms the baseline in terms of memorization capability. The experimental results are summarized in the table below:
>
> | Method\Metric | SSIM↑ | PSNR↑ | LPIPS↓ | FVD↓ |
> | --- | --- | --- | --- | --- |
> | DFoT | 0.4436 | 13.03 | 0.4469 | 337.5 |
> | VRAG | **0.9116** | **32.21** | **0.1146** | **221** |
>
> We believe these results effectively demonstrate the generalizability of our approach beyond Minecraft.
>
> > W3: The citations are totally off in this paper, which makes the positioning very strange. In the opening statement, the authors say “Foundational world models capable of simulating future outcomes based on different actions 21 are crucial for effective planning and decision making [1, 2, 3].” and then cite [1] a NeurIPS 2015 paper which is not a foundation world model, [2] David Ha World Models paper, which is iconic, but also not a foundation world model, and then [3] Dreamer, which is a single environment, not foundation world model. A foundation world model is one where it can be prompted to generate new worlds, which none of these can. The only foundation world models produced so far are Genie1, Genie2 and Cosmos, as far as I am aware.
> >
>
> **Reply to W3:** We appreciate the reviewer for pointing out misleading references for “foundational world models” and will modify accordingly.
>
> > W4: The authors incorrectly reference the Genie 2 model “One line of research addressing this involves predicting the underlying 3D world structure like Genie2”. This is incorrect, Genie 2 was a latent diffusion model, with no inductive bias. If anything, it is very close related work.
> >
>
> **Reply to W4:** Here we aim the emphasize the distinguished 3D world modeling capability of Genie2 compared with Genie, which only models 2D visual features with much less spatiotemporal consistency. We didn’t indicate the inductive bias in the method, but will modify this sentence to avoid any confusion.
>
> > Q1: Do you have any fair comparisons w.r.t memory usage for your method vs. baselines? I assume yours uses more, for the same model size, so it is not a fair comparison.
> >
>
> **Reply to Q1:** All models are trained on the same dataset. The baselines DF20 and History Buffer exhibit comparable memory usage and inference-time overhead to our method during both inference and training phases. The inference time is derived for autoregressive generation over 600 frames. DF10 demonstrates lower inference-time overhead due to shorter context, YaRN incurs higher overhead. We think that this constitutes a fair and balanced comparison.
>
> | Method | DF10 | DF20 | YaRN | VRAG |
> | --- | --- | --- | --- | --- |
> | Context length (Frame) | 10 | 20 | 40 | 20 |
> | Memory usage (MB) | 4420 | 4448 | 4543 | 4452 |
> | Inference Time (min) | 9 | 12 | 23 | 12 |
>
> > Q2: What if you scale up the baseline?
> >
>
> **Reply to Q2:** We sincerely would like the reviewer to clarify which aspect of "scale up" that is referred to. To alleviate the concerns, we show results with longer context and more training data.  First, we increased the window size to 40 (DF40) on the same dataset. Second, we trained DF20 on a significantly larger dataset—collecting 6x more data (100 hours of video), which required 4 days of training on 8 A100 GPUs. The experimental results are as follows:
>
> | Method\Metric | SSIM↑ | PSNR↑ | LPIPS↓ |
> | --- | --- | --- | --- |
> | DF20 | 0.466 | 16.643 | 0.538 |
> | DF40 | 0.484 | 17.048 | 0.500 |
> | DF20 (Larger dataset) | 0.482 | 16.908 | **0.496** |
> | VRAG | **0.506** | **17.097** | 0.506 |
>
> As shown in the table, increasing the dataset size or window size does help improve the model's performance, but it still cannot fully resolve the issue of compounding errors and insufficient memory. VRAG still outperforms in both SSIM and PSNR metrics against the baselines with larger scales, demonstrating the effectiveness of our method.
>
> **Reference**
>
> [1] History-Guided Video Diffusion, Arxiv 2025

---

> > ### Comment · Area_Chair_Ja1Z · 2025-08-05
> >
> > Reviewer udTT, can you please check whether the author's rebuttal addresses your concerns?

---

> > ### Comment · Reviewer_udTT · 2025-08-05
> > **Keeping score**
> >
> > Hi - thank you for the rebuttal. I don't have too much to say, since overall my score is remaining the same. I respect that the authors have tried to contribute to an exciting field, but I do not believe the insights from this work will translate to larger scale systems, given the tiny dataset used. I appreciate the additional "larger scale" experiments finetuning the diffusion forcing model, but for that to be a fair comparison we would need a FLOPs neutral memory neutral baseline which we don't have in the rebuttal.

---

### Note · Authors · 2025-08-12

Dear ACs, SACs, PCs, and Reviewers:

Thank you for your dedication to this conference. As the review process nears completion, please allow us to summarize the key points of this review and express our sincere gratitude to all participants.

We are pleased to have received several high-quality reviews, which we have promptly addressed and utilized to make corresponding improvements to our manuscript. The reviewer praised our idea to be neat and novel (Reviewer udTT), and noted that it includes a thorough ablation study (Reviewer UWPa). They believe our approach is a promising direction and expressed appreciation for our idea of history retrieval (Reviewer M9fd).

After the Author/Reviewer Discussion, our work has received positive appreciation from all reviewers. The reviewers’ attitudes towards our work have shifted more positively, reflecting a endorsement of its value. The reviewers have expressed recognition of our responses:

> Reviewer udTT commented: "I respect that the authors have tried to contribute to an exciting field, but I do not believe the insights from this work will translate to larger scale systems, given the tiny dataset used."

> Reviewer UWPa commented: "All my conerns are well explained."

> Reviewer M9fd commented: "Thanks for your additional experiments to address my concerns. I will increase my score accordingly."

> Reviewer 4P6H commented: "All my concerns have been addressed."

Note that Reviewer udTT still concern about dataset size, we would like to clarify that our goal in this work is to validate the feasibility of our method in the Minecraft setting, rather than building a foundation-level world model. As mentioned in the rebuttal, we have already expanded the baseline dataset (6× larger than the original setting) and conducted extensive training for 4 days on 8 A100 GPUs—a significant computational expense for academic research. While we regret that we lack the resources to train on ultra-large-scale datasets, we believe the current experiments sufficiently demonstrate the effectiveness of our approach within the intended scope.

Finally, we would like to express our gratitude once again for your contribution to this conference, enabling us to benefit from the valuable insights of the reviewers and improve our paper. We sincerely hope that you will appreciate our work and support its better exposure.

Sincerely,

Authors of the paper 22957

---

### Decision · Program_Chairs · 2025-09-17

**Decision:**

Accept (poster)

**Comment:**

This paper introduces Video Retrieval Augmented Generation (VRAG),    to improve long-term consistency and mitigate compounding errors in interactive video generation. Initially, all reviewers expressed significant concerns about the limited evaluation, which was confined to a small Minecraft dataset, raising questions about generalizability and the suitability of the evaluation metrics. However, the authors provided a thorough rebuttal that included new experiments on a real-world dataset and additional metrics for temporal consistency, which successfully addressed the majority of the issues raised. While one reviewer remained concerned that the insights gained from the small-scale experiments might not translate to larger systems, the overall consensus is in favor of the paper. The proposed method is a neat and valuable contribution.